# Modular control of vertebrate axis segmentation in time and space

Ali Seleit [ID][1], Ian Brettell[2], Tomas Fitzgerald[2], Carina Vibe [ID][1], Felix Loosli[3], Joachim Wittbrodt [ID][4], Kiyoshi Naruse[5], Ewan Birney [ID][2✉] & Alexander Aulehla [ID][1✉]

## Abstract

**How the timing of development is linked to organismal size is a longstanding question. Although numerous studies have reported a correlation of temporal and spatial traits, the developmental or selective constraints underlying this link remain largely unexplored. We address this question by studying the periodic process of embryonic axis segmentation in-vivo in *Oryzias* fish. Interspecies comparisons reveal that the timing of segmentation correlates to segment, tissue and organismal size. Segment size in turn scales according to tissue and organism size. To probe for underlying causes, we genetically hybridised two closely related species. Quantitative analysis in ~600 phenotypically diverse F2 embryos reveals a decoupling of timing from size control, while spatial scaling is preserved. Using developmental quantitative trait loci (*dev*QTL) mapping we identify distinct genetic loci linked to either the control of segmentation timing or tissue size. This study demonstrates that a developmental constraint mechanism underlies spatial scaling of axis segmentation, while its spatial and temporal control are dissociable modules.**

**Keywords** Developmental Timing; QTL Mapping; Inter-species Hybridization; Scaling; Somitogenesis
**Subject Categories** Development; Evolution & Ecology

## Introduction

How developmental timing is controlled and linked to the form and function of developing organisms is a longstanding, fundamental question. Comparative studies in a large number of phyla have revealed a correlation between organismal size and developmental timing (Gillooly and Dodson, 2000; Pauly and Pullin, 1988; Peters, 1983; Berrill 1935). The repeated, albeit certainly not universal (Church et al, 2019), documentation of scaling relationships between timing and size have fueled decades long discussions about possible underlying fundamental constraints and scaling laws (Gillooly et al, 2002; Peters, 1983). However, the lack of clear insight into the nature and origin of the underlying mechanisms, i.e. developmental constraints vs. selective pressures, have precluded a deeper understanding of the implications of observed correlations. The challenge of tackling questions related to the timing of development are also due to the fact that timing is under complex control, integrating both genetic (Harima et al, 2013; Herrgen et al, 2010; Liao et al, 2016) and environmental factors (Schröter et al, 2008; Vibe, 2020), which combined result in a species-characteristic timing. For instance, in vertebrates, the completion of body axis segmentation into the pre-vertebrae takes ~15 days in humans (O'Rahilly and Müller, 2003; Schoenwolf et al, 2021), ~4 days in mice (Tam, 1981; Theiler, 1989), ~3 days in chick (Bénazéraf and Pourquié, 2013; Hamburger and Hamilton, 1992) and ~1 day in zebrafish embryos (Kimmel et al, 1995; Schröter et al, 2008). The species-characteristic timing of body axis segmentation is linked to the underlying activity of the segmentation clock (Palmeirim et al, 1997). The clock activity can be quantified at the level of ultradian Notch-signaling oscillations that occur in presomitic mesoderm (PSM) cells (Lauschke et al, 2013) with a period matching the species-characteristic rate of addition of somites, ~6 h in human (Miao et al, 2023; Sanaki-Matsumiya et al, 2022), ~2 h in mouse (Aulehla et al, 2008) ~90 min in chick (Palmeirim et al, 1997) and ~30 min in zebrafish embryos (Soroldoni et al, 2014). How differences in timing relate to distinct morphologies and proportions is challenging to address in evolutionarily distant species (Diaz-Cuadros et al, 2023; Lázaro et al, 2023; Matsuda et al, 2020). We hence employed a comparative, *common garden* (De Villemereuil et al, 2016) approach using closely related medaka fish species, *Oryzias sakaizumii* and *latipes*, that are native to different regions in Japan. The northern *Oryzias sakaizumii* (Kaga, HNI) and southern *Oryzias latipes* (Cab, HdrII, Ho5) have an estimated evolutionary divergence time of ~18 million years (Sakaizumi, 1984; Setiamarga et al, 2009). Importantly, these species exhibit developmental and phenotypic variation (Asai et al, 2011; Katsumura et al, 2019; Kinoshita et al, 2009) tolerance to inbreeding (Fitzgerald et al, 2022; Kinoshita et al, 2009) and are amenable to interbreeding. We developed a real-time imaging approach, which enables a quantitative analysis of both the temporal and spatial measures of embryonic axis segmentation in genetically diverse F2 offspring resulting from

¹Developmental Biology Unit, European Molecular Biology Laboratory, Heidelberg, Meyerhofstrasse 1, 69117 Heidelberg, Germany. ²European Molecular Biology Laboratory, Wellcome Genome Campus, Hinxton, Cambridge, UK. ³Institute of Biological and Chemical Systems, Karlsruhe Institute of Technology, Hermann-von-Helmholtz-Platz 1, 76344 Eggenstein-Leopoldshafen, Karlsruhe, Germany. ⁴Centre for Organismal Studies, Ruprecht Karls Universität Heidelberg, Im Neuenheimer Feld 230, 69120 Heidelberg, Germany. ⁵National Institute for Basic Biology, Nishigonaka 38, Myodaiji, Okazaki 444-8585 Aichi, Japan. ✉E-mail: birney@ebi.ac.uk; aulehla@embl.de

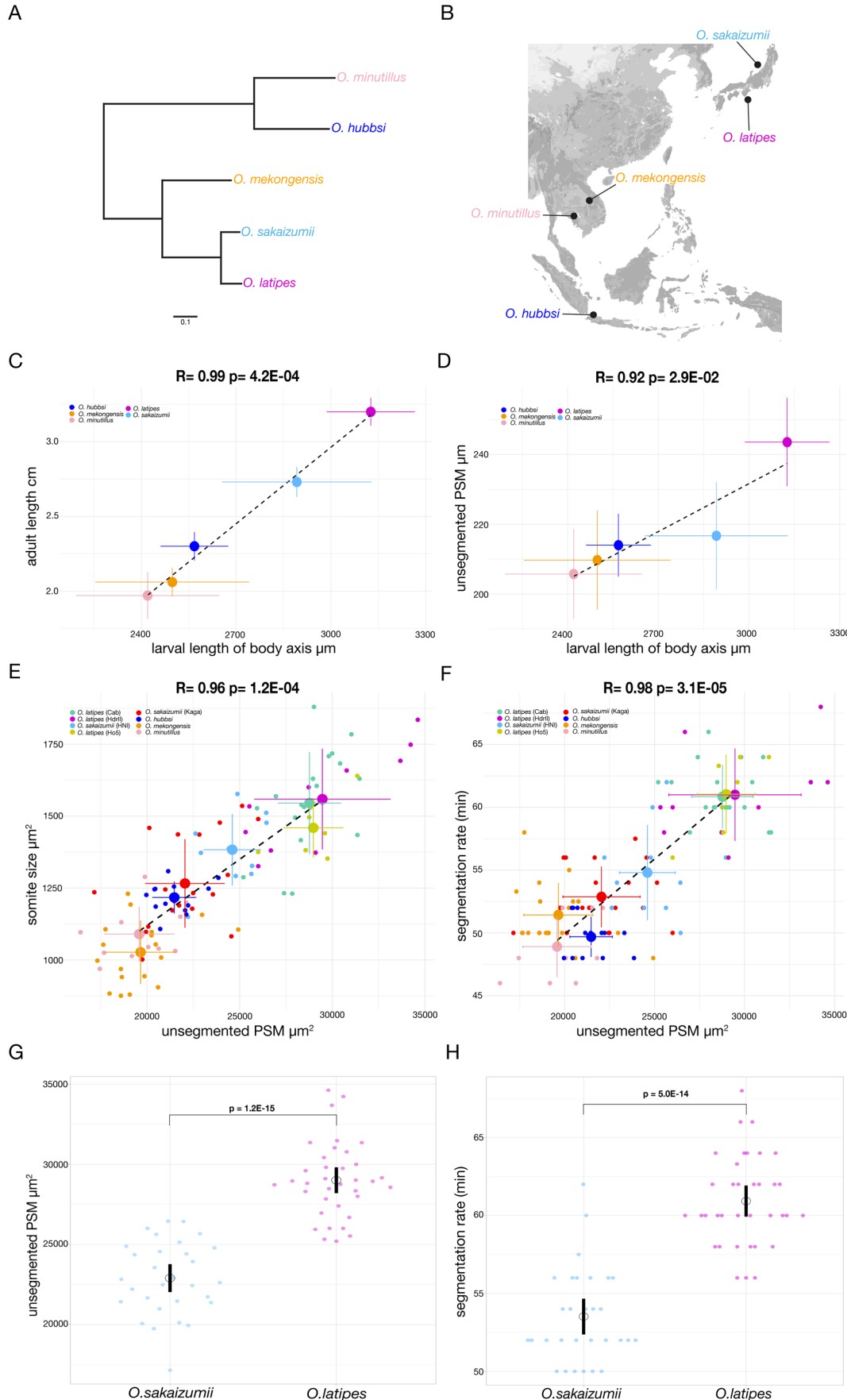

◄  **Figure 1.  Scaling of segmentation timing and size in the *Oryzias* genus.**

(A) Phylogenetic tree of *Oryzias* species *O. minutillus* (pink), *O. hubbsi* (blue), *O. mekongensis* (orange), *O. sakaizumii* (light blue) and *O. latipes* (magenta). (B) Geographical locales across south-east Asia of *Oryzias* species assayed in this study. (C) Pearson's correlation on average trait values for larval length of segmented body axis (stage 42) and adult length (11 months old) in *O. minutillus* (pink), *O. hubbsi* (blue), *O. mekongensis* (orange), *O. sakaizumii* (light blue) and *O. latipes* (magenta) shows a positive correlation $R = 0.99$ $P$ value $= 4.2E\text{-}04$. Black dotted line$=$ linear fit on average trait values, vertical and horizontal lines on both axes represent standard deviation (SD). For adult length $N = 10$ fish *O. mekongensis*, *O. minutillus*, $N = 12$ fish *O. hubbsi*, $N = 13$ fish *O. sakaizumii*, $N = 15$ fish *O. latipes*. For larval length $N = 10$ larvae *O. hubbsi*, *O. mekongensis*, $N = 11$ larvae *O. minutillus*, $N = 30$ larvae *O. latipes*, $N = 20$ larvae *O. sakaizumii*. (D) Pearson's correlation on average trait values for larval length of segmented body axis (stage 42) and unsegmented PSM length (10SS) *O. minutillus* (pink), *O. hubbsi* (blue), *O. mekongensis* (orange), *O. sakaizumii* (light blue) and *O. latipes* (magenta) shows a positive correlation $R = 0.92$ $P$ value $= 2.9E\text{-}02$. Black dotted line$=$ linear fit on average trait values, vertical and horizontal lines on both axes represent standard deviation (SD) For larval length $N = 10$ larvae *O. hubbsi*, *O. mekongensis*, $N = 11$ larvae *O. minutillus*, $N = 30$ larvae *O. latipes*, $N = 20$ larvae *O. sakaizumii* For PSM length $N = 13$ *O. hubbsi*, $N = 18$ *O. mekongensis*, $N = 11$ larvae *O. minutillus*, $N = 36$ larvae *O. latipes*, $N = 30$ *O. sakaizumii*. (E) Pearson's correlation on average trait values of unsegmented presomitic mesoderm (PSM) size and nascent somite size at 10-11SS in *O. minutillus* (pink), *O. hubbsi* (blue), *O. mekongensis* (orange), *O. sakaizumii* Kaga (red), HNI (light blue) and *O. latipes* Cab (green), HdrII (magenta), Ho5 (yellow) embryos shows a positive correlation $R = 0.96$ $P$ value $= 1.2E\text{-}04$. Black dotted line$=$ linear fit on average trait values, vertical and horizontal lines on both axes represent standard deviation (SD). Individual data points are shown for each population $N = 11$ *O. minutillus*, $N = 13$ *O. hubbsi*, $N = 18$ *O. mekongensis*, $N = 19$ Cab, $N = 10$ HdrII, $N = 7$ Ho5, $N = 20$ Kaga, $N = 10$ HNI. (F) Pearson's correlation on average trait values of unsegmented presomitic mesoderm (PSM) size and segmentation rate at the 10-11SS in *O. minutillus* (pink), *O. hubbsi* (blue), *O. mekongensis* (orange), *Oryzias sakaizumii* Kaga (red), HNI (light blue) and *Oryzias latipes* Cab (green), HdrII (magenta), Ho5 (yellow) embryos shows a positive correlation $R = 0.98$ $P$ value $= 3.1E\text{-}05$. Black dotted line$=$ linear fit. vertical and horizontal lines on both axes represent standard deviation (SD). Individual data points are shown for each population $N = 11$ *O. minutillus*, $N = 13$ *O.hubbsi*, $N = 18$ *O. mekongensis*, $N = 19$ Cab, $N = 10$ HdrII, $N = 7$ Ho5, $N = 20$ Kaga, $N = 10$ HNI. (G) unsegmented presomitic mesoderm (PSM) size at the 10-11 somite stage (SS) obtained from brightfield live-imaging. *Oryzias sakaizumii* have a smaller PSM size compared to *Oryzias latipes* (22,898 $\mu m^2$ (SD $+/-2293$) vs. 29,004 $\mu m^2$ (SD $+/-2350$)). Black circle $=$ mean period. Black line $=$ 95% confidence interval. Welch two sample $t$ test $P = 1.2E\text{-}15$. $N = 30$ *Oryzias sakaizumii*, $N = 36$ *Oryzias latipes*. (H) Embryonic segmentation rate at the 10-11 somite stage (SS) obtained from brightfield live-imaging. *Oryzias sakaizumii* have a faster axis segmentation rate compared to *Oryzias latipes* (53.52 min (SD $+/-3.0$) vs. 60.92 min (SD $+/-2.91$)). Black circle $=$ mean period. Black line $=$ 95% confidence interval. Welch two sample $t$ test $P = 5.0E\text{-}14$ $N = 30$ *Oryzias sakaizumii*, $N = 36$ *Oryzias latipes*. Source data are available online for this figure.

interbreeding of *O. sakaizumii* and *O. latipes*. We combined the phenotypic analysis with whole genome sequencing of ~600 F2 embryos and performed *developmental* quantitative trait loci (*dev*QTL) mapping to gain insight into how the control of *time* and *space* is functionally linked during development.

# Results and discussion

## Correlation of segmentation timing and size in the *Oryzias* genus

To investigate the relationship between developmental timing and size we first assayed several *Oryzias* species covering a broad geographical range in south-east Asia and spanning an evolutionary time of 50 million years (Yamahira et al, 2021) (Fig. 1A,B). Our quantifications showed a strong correlation between adult, larval and embryonic sizes (Figs. 1C,D and EV1A,B). In addition, embryos belonging to species with larger presomitic mesoderm (PSM) formed proportionally larger segments (Figs. 1E and EV1C–G). As seen previously at the organismal level, we indeed found that the process of axis segmentation showed a correlation at the level of spatial and temporal control - faster axis segmentation occurred in embryos of smaller size (Figs. 1F and EV1B). We included a correction for phylogenetic relatedness (Felsenstein, 1985; Symonds and Blomberg, 2014) to solidify this result ("Methods"). Taken together these results revealed a correlation of *temporal* and *spatial* measures during the embryonic axis segmentation process. We then investigate the underlying mechanisms and asked, whether these correlations reflect a single developmental process and constraint or rather, multiple developmental modules that become coupled based on other, selective pressures. To address this question, we exploited the ability to interbreed and produce fertile offspring using southern *O. latipes* and northern *O. sakaizumii*, two species exhibiting clear differences in size and developmental timing (Figs. 1G,H and EV1H,I).

## Segmentation clock period in north/south hybrid F1 embryos

Based on our initial findings that revealed differences in timing and morphology between northern and southern species of medaka, we performed interbreeding experiments to analyse the degree of heritable variation in F1 and F2 offspring. We first analysed hybrid F1 offspring from a series of north-south crosses (Fig. 2A). We used in-vivo live-imaging to quantify timing of axis segmentation in hybrid F1 embryos at the 10-11 somite stage (SS) (Fig. 2B–E). To enable a precise quantification of timing differences we made use of a fluorescent segmentation clock endogenous knock-in *her7-venus* reporter line that we recently developed (11) (Fig. 2B,C; Fig. S1A,B; Data EV1). Our results revealed that the segmentation clock period differed across the hybrid F1s (Fig. 2D), with the *Oryzias sakaizumii/latipes* hybrid F1 embryos (HNI/Cab, Kaga/Cab) showing a faster segmentation timing than hybrid F1 *Oryzias latipes/latipes* (Ho5/Cab, HdrII/Cab) (Fig. 2D,E; Fig. S1C,D). To address the possible impact of maternal effects, we performed reciprocal (north/south) Kaga/Cab F1 crosses. The results showed that while egg size is maternally controlled as expected, neither PSM size nor segmentation clock period measurements showed evidence for a significant maternal effect (Fig. EV2A–I). Interestingly, PSM size in the F1 reciprocal cross did show a size more comparable to Cab than Kaga (Fig. EV2F). Next, we generated F2 offspring by crossing hybrid F1 Kaga/Cab (north/south) fishes, with the goal of producing individuals having a unique genetic composition following meiotic recombination events.

## Modular control of segmentation timing and size

We quantified the segmentation clock period in 638 F2 embryos and found a wide distribution of timings that exceeded those occurring in the parental populations (Fig. 3A; Fig. S2A–C). The statistical analysis of variances in the parental *vis-a-vis* F2 samples

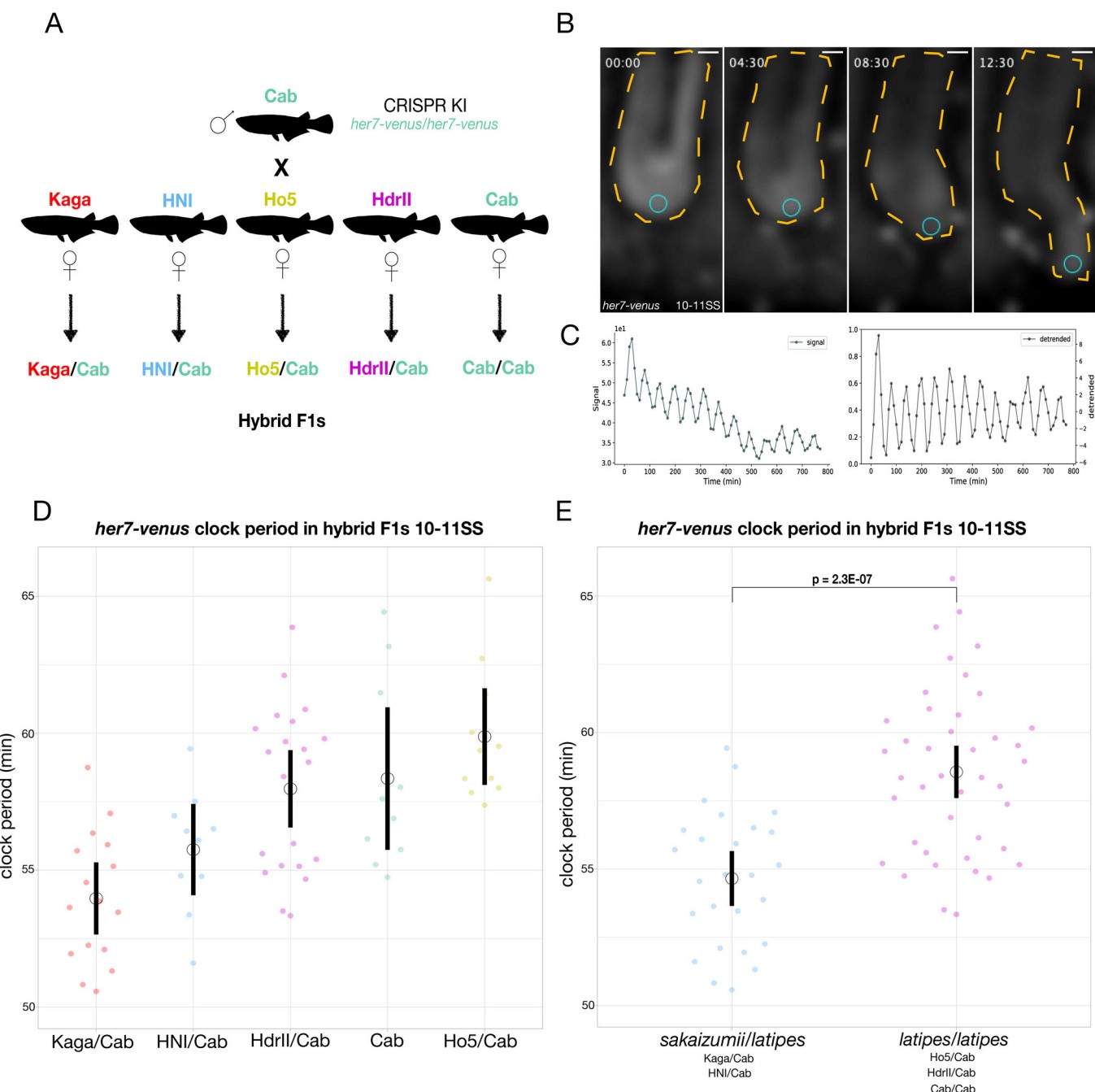

**Figure 2. Hybrid north/south F1 embryos show a faster segmentation clock period than hybrid south/south F1 embryos.**

(**A**) Schematic diagram of genetic crosses performed to generate hybrid F1 fish. A male Cab homozygous for the endogenously tagged clock oscillator *her7-venus* is crossed to *wt* Kaga, HNI, Ho5 HdrII and Cab females. (**B**) Selected frames from in-vivo tail imaging of endogenous *her7-venus* oscillations at the 10–11 somite stage. Blue circle indicates the location of extracted raw intensity grey values for posterior period analysis over the course of imaging. Yellow dotted lines highlight the outlines of the tail tissue. Time in hours. Scale bar = 30 μm. (**C**) Raw and detrended signal graphs extracted from the mean intensity grey values of *her7-venus* expression in (**B**) show oscillatory signal. (**D**) endogenous *her7-venus* clock period measurements in hybrid F1 embryos at the 10-11SS. Kaga/Cab F1 hybrids have the fastest *her7-venus* clock period (53.97 min (SD +/− 2.40)), while HNI/Cab F1s show (55.75 min (SD +/− 2.21)), hybrid F1 Cab/Cab show (58.34 min (SD +/− 3.45)), Cab/HdrII show (57.97 min (SD +/− 3.24)), Cab/Ho5 show (59.88 min (SD +/− 2.51)). One-way ANOVA P = 2.2E-05. Post-Hoc Tukey HSD testing shows significant differences between the following groups Kaga/Cab and Cab/Cab P adjusted = 2.2E-03, Kaga/Cab and HdrII/Cab P adjusted = 4.0E-04, Kaga/Cab and Ho5/Cab p adjusted = 8.9E-06, HNI/Cab and Ho5/Cab p adjusted = 9.7E-03. N = 16 Kaga/Cab F1, N = 10 HNI/Cab F1, N = 10 Cab/Cab F1, N = 21 HdrII/Cab F1, N = 11 Ho5/Cab F1. Black bars indicate 95% confidence interval, Black circle indicates the mean. (**E**) endogenous *her7-venus* clock period measurements in hybrid F1 of *Oryzias sakaizumii/Oryzias latipes* and *Oryzias latipes/Oryzias latipes* embryos at the 10-11SS. *Oryzias sakaizumii/Oryzias latipes* F1 hybrids have a faster *her7-venus* clock period (54.65 min (SD +/− 2.45)) than hybrid *Oryzias latipes/Oryzias latipes* embryos (58.56 min (SD +/− 3.05)). Welch two sample *t* test P = 2.3E-07. N = 26 *Oryzias sakaizumii/Oryzias latipes* F1 hybrid embryos, N = 42 *Oryzias latipes/Oryzias latipes* F1 hybrid embryos. Black bars indicate 95% confidence interval, Black circle indicates the mean. Source data are available online for this figure.

indicated equality (F-test for equality of variances $P = 0.42$ and 0.11) and therefore the wide distribution we report in the F2s is evidence of transgressive segregation (Rieseberg et al, 1999, 2003). The distribution of segmentation clock periods in the F2 offspring was continuous from the fastest (47.1 min) to the slowest (69.2 min), showing a 22.1 min period difference (Fig. 3A; Fig. S2D). In addition, we measured PSM and nascent somite size in the F2 embryos, and like segmentation timing, we found a continuous distribution and a wide phenotypic range that exceeded the parental extremes in both directions (Fig. 3B,C; Fig. S2E,F). Taken together the data obtained from this F2 cross argues for the polygenic nature and complex genetic control of segmentation timing, PSM size and somite size. Interestingly, while further analysing the considerable variation in spatial measures observed in F2 embryos, we found a correlation of nascent somite size to PSM size that mirrored the spatial scaling found across the five different inbred strains (Fig. 3D). The maintenance of a linear relationship between PSM and somite size even in F2 embryos provides evidence for a developmental constraint mechanism (Alberch, 1989; Smith et al, 1985) underlying segment scaling. Importantly, however, we found that the correlation between PSM/nascent somite size and segmentation clock period, which we had seen across the *Oryzias* species, is absent in the F2 data (Fig. 3E,F; Fig. S2G–L). And while the correlation between vertebral count and larval length that we find in the *Oryzias* species is present in the F2 cross (Fig. EV3A–K), the correlation between segmentation clock period and both vertebral count and larval length is absent in the F2 data (Fig. EV3G–I). These results indicate that segmentation timing and size can be functionally decoupled.

To link the phenotypes to the underlying genetics we performed *developmental*QTL mapping in F2 embryos. This was based on whole genome sequencing (WGS) of the parental Cab and Kaga fish with high coverage (26x and 29x, respectively) (Fig. 3G–I; Fig. S3A). We observed a high level of homozygosity (83% of all loci genome-wide) in Cab, while in Kaga, homozygosity was lower (31% homozygosity across all loci genome-wide) (Fig. 3H,I). In total, we identified ~2.2 million homozygous divergent SNPs between Kaga and Cab that segregated as expected in the F1 Kaga/Cab hybrids (Fig. S3B–D). To uncover the genetic basis underlying segmentation timing and size control we built a genome-level genetic recombination map for every F2 embryo by whole genome sequencing (WGS) of 600 F2 embryos with low coverage (~ 1×) (Fig. 3J). In conjunction with deep sequencing of the parental populations and hybrid F1s (Fig. 3H,I; Fig. S3A–D) we were able to assign one of three genotypes (homozygous Cab, heterozygous Cab/Kaga, homozygous Kaga) to every genomic position for each F2 embryo (Fig. 3J; Fig. S3E,F, "Methods"). Combining our quantitative phenotypic measurements with the genomic information of individual F2 embryos we performed *developmental* QTL (*dev*QTL) mapping on both traits.

## *dev*QTL mapping and functional validation on segmentation timing and PSM size

We used the Genome-Wide Complex Trait Analysis (GCTA) (Yang et al, 2011) implementation of a linear mixed model to map the *dev*QTLs in F2 embryos associated with segmentation timing ("Methods"). This revealed several genomic loci that passed the significance threshold, located on chromosomes 3, 4 and 10

(Fig. 4A). These regions contained a total of 46,872 single nucleotide polymorphisms (SNPs) that were homozygous-divergent in the F0 parental strains (Cab and Kaga), the majority of which occurred in non-coding or intergenic portions of the genome (Fig. EV4A,B). Overall, the SNPs fall within the genomic coordinates of 57 genes (Data EV2). To further refine our search for candidates, we assayed which of the genes are transcriptionally active within the PSM, using bulk-RNA sequencing on *O.sakaizumii* and *O.latipes* tails (Fig. EV4C,D; Data EV3). We found 35 out of 57 genes to be expressed in the PSM, five of those showed differential expression between the two species. We next categorized these genes into differentially expressed transcripts between *O. sakaizumii* and *O. latipes* ($n = 5$) and a second group consisting of genes harboring one or more SNPs with a coding consequence ($n = 29$). For in-vivo functional testing we selected all differentially expressed genes in addition to six candidates harboring only a coding consequence based on GO analysis (Ge et al, 2020) (Fig. EV4E). We employed an F0 CRISPR/Cas9 knock-out approach (Hoshijima et al, 2019; Kroll et al, 2021; Seleit et al, 2020; Wu et al, 2018) and quantified segmentation clock period using *her7-venus* in-vivo imaging (Figs. 4B and EV4F–H; Fig. S4A–D). Two out of eleven targeted genes, i.e. *mesoderm posterior b* (*mespb*) on chromosome 3 and *paraxial protocadherin 10b* (*pcdh10b*) on chromosome 10, showed a minor but significant decrease of segmentation clock period (Figs. 4B and EV4F–H; Fig. S4A–D). Hybridisation chain reaction (HCR) on both genes in the parental Kaga and Cab strains showed expression domains in the unsegmented PSM (Fig. S4E–H"). To assess whether the effects of both genes are additive, we performed combinatorial targeting of *mespb/pchd10b* (Fig. 4B; Fig. S4A). Our results showed a phenocopy of the single knock-outs, suggesting the effects are non-additive and could be mediated through a common genetic pathway. However, at the 10-11 somite stage *mespb* and *pchd10b* show complementary expression domains in the PSM (Fig. S4E–H"), raising the possibility of an indirect/earlier genetic interaction. To investigate this further we performed bulk RNA-sequencing on Cab *wild-type* and *mespb* mutant PSMs, this revealed differential expression of 808 genes (Fig. EV5A and Data EV4), interestingly many of the downregulated transcripts include posteriorly expressed genes (e.g *tbxt, foxb1, fgf8, axin2*) in agreement with an indirect/earlier genetic effect (Fig. EV5A and Data EV4).

The *dev*QTL mapping using PSM size as a linked trait identified a single significant region on chromosome 3 (Fig. 4C). This region is distinct from the one we obtained on the same chromosome for segmentation timing. In this region, we identified 204 genes, of which 155 are expressed in the PSM (Fig. EV5B,C; Data EV5). As a proof of principle, we selected 4 candidate genes based on GO annotation and performed CRISPR/Cas9 loss-of-function analysis in F0 embryos. We were able to identify two Crispants, i.e. *atxn1l* (a notch co-factor) and *dll1* (a notch ligand), that showed a reduction in PSM size compared to control embryos (Figs. 4D and EV5D-G'; Fig. S5A–C; Data EV6). In both *atxn1l* and *dll1* Crispants we assayed whether there was an effect on segmentation timing. We were able to extract reliable segmentation period measurements only from a subset of *atxn1l* and *dll1* Crispant embryos, likely due to an overall downregulation of *her7* expression, however, the results showed that segmentation period does not differ from control embryos for either Crispant (Fig. S5D–H), despite a

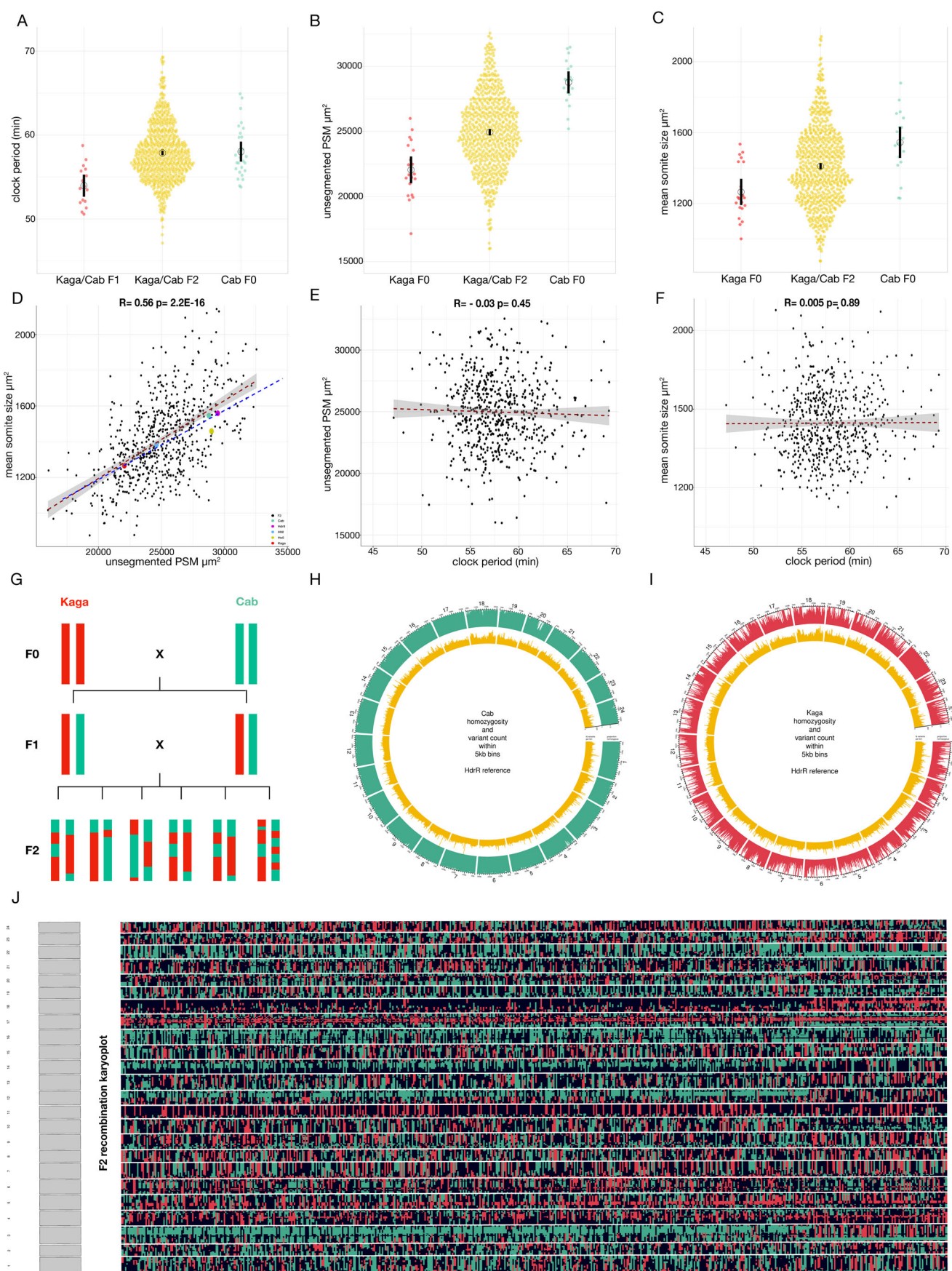

**Figure 3.  Genotype-phenotype map of segmentation timing and size in an F2 Kaga/Cab cross.**

(A) endogenous *her7-venus* clock period measurements in F2 Kaga/Cab embryos (57.86 min (SD +/− 3.43)), F1 Kaga/Cab (53.97 min (SD +/− 2.4)) and F0 Cab/Cab (58.03 min (SD +/− 3.02)). Quantifications were done at the 10-11SS. Black circle = mean period. Black line = 95% confidence interval. Each dot is one embryo. $N = 638$ F2 Kaga/Cab embryos $N = 16$ F1 Kaga/Cab embryos $N = 28$ F0 Cab embryos. (B) unsegmented PSM size in F2 Kaga/Cab (mean = 24,944 $\mu m^2$ (SD +/− 2930)) as compared to the parental F0 Cab (mean = 28,770 $\mu m^2$ (SD +/− 1709)) and Kaga (mean = 22,048 $\mu m^2$ (SD +/− 2145)). $N = 19$ Cab, $N = 20$ Kaga, $N = 633$ Kaga/Cab F2 (C) mean somite area of nascent somites in Kaga and Cab F0 embryos at the 10-11SS compared to the F2 Kaga/Cab cross. Average sizes: F2 Kaga/Cab (1411 $\mu m^2$ (SD +/− 222)), Kaga (1265 $\mu m^2$ (SD +/− 154)), Cab (1545 $\mu m^2$ (SD +/− 177)). $N = 20$ Kaga F0 embryos, $N = 19$ Cab F0 embryos, $N = 631$ Kaga/Cab F2 embryos. (D) Pearson's correlation of mean somite size and unsegmented PSM size across all F2 Kaga/Cab embryos $R = 0.56$ $P$ value = 2.2E-16. Red dotted line= linear fit for F2 data, grey shaded area= 95% confidence interval for F2 data, slope = 0.044. Blue dotted line= linear fit for F0 average data (Cab, HdrII, HNI, Ho5, Kaga, coloured dots), slope= 0.046 $N = 631$ Kaga/Cab F2 (black dots). (E) Pearson's correlation between clock period and unsegmented PSM size in F2 Kaga/Cab cross $R = -0.03$ $P$ value = 0.45, $N = 623$ Kaga/Cab F2. (F) Pearson's correlation between mean somite size and clock period across all F2 Kaga/Cab embryos $R = 0.005$ $P$ value = 0.89. Red dotted line= linear fit, grey shaded area= 95% confidence interval $N = 623$ Kaga/Cab F2. (G) schematic diagram of genetic crosses to generate F2 Kaga/Cab embryos showing only one chromosome and assuming homozygosity across all sites. Crossing Kaga F0 to Cab F0 generates a hybrid F1 that is heterozygous at all sites. Incrossing F1 Kaga/Cab hybrids generates an F2 population with each individual having a unique genetic composition due to the random nature of meiotic recombination events. (H) Circos plot showing the 24 chromosomes of whole genome sequenced F0 Cab embryos aligned against the HdrII reference genome for southern medaka populations. Proportion of homozygous SNPs within 5 kb bins in the Cab F0 genome is shown in green and number of SNPs in each bin in yellow. The mean homozygosity across all bins is 83%. (I) Circos plot showing the 24 chromosomes of a whole genome sequenced F0 Kaga embryo aligned against the HdrII reference genome. Proportion of homozygous SNPs within 5 kb bins in the Kaga F0 genome is shown in red and number of SNPs in each bin in yellow. The mean homozygosity across all bins is 31%. (J) Recombination karyoplot for all 24 chromosomes of whole genome sequenced F2 Kaga/Cab embryos. The plot is based on the ratio of reads mapping to either the Cab or Kaga allele within 5-kb bins (details in "Methods"). Homozygous Cab blocks are shown in green, heterozygous loci are shown in black and homozygous Kaga loci are shown in red. $N = 600$ F2 Kaga/Cab embryos. Source data are available online for this figure.

reduction in PSM size. Relatedly, we found that in segmentation timing mutants PSM size is either unaffected (*pcdh10b* mutants) or slightly reduced (*mespb* mutants) compared to *wild-types* (Fig. EV5H–H'). The CRISPR/Cas9 knock-out approach provides proof of principle validation that the *dev*QTL mapping identified functionally relevant genomic regions linked to the control of segmentation timing and PSM size, respectively.

In this study we were able to reveal, using genetic crosses of closely related medaka species, evidence for a developmental constraint mechanism underlying segment size scaling, forming a single functional module. In contrast, we found clear evidence that the timing and spatial control of axis segmentation can be decoupled and hence are distinct modules. Interestingly, our data argues that these modules are coupled in the natural context, as we reveal a correlation of spatial and temporal developmental traits in the interspecies comparison across the *Oryzias* genus. It is therefore possible that selective pressures are linking these modules thereby restricting the phenotypic outcomes realized in the natural setting. Indeed, the ecological niches of *O. sakaizumii* and *O.latipes* are known to differ with northern strains experiencing lower temperatures, a shorter breeding season and having a faster juvenile growth rate than southern strains (Asai et al, 2011; Katsumura et al, 2019; Kinoshita et al, 2009; Setiamarga et al, 2009). Future investigations will therefore aim to understand how evolutionary pressures and developmental modules are integrated, taking into account the specific environmental contexts and life-history trade-offs involved.

## Methods

### Animal husbandry and ethics

Medaka *Oryzias latipes* (Cab, HdrII, Ho5), *Oryzias sakaizumii* (HNI and Kaga) strains (Iwamatsu, 2004; Kasahara et al, 2007; Naruse et al, 2004), *her7-venus* Cab (Vibe, 2020), *Oryzias mekongensis*, *Oryzias hubbsi*, and *Oryzias minutillus* were maintained as closed stocks in a constant recirculating system at 27–28 °C, with a 14 hr light/10 h dark cycle in the EMBL Laboratory Animal Resources (LAR) fish facility.

Both Males and females were used for experiments. Animal experiments were performed after project approval by the EMBL Institutional Animal Care and Use Committee (IACUC), IACUC project code is 20/001_HD_AA.

### Live-imaging sample preparation

Embryos were prepared for live-imaging as previously described (Seleit et al, 2017; Seleit et al, 2017). 1× Tricaine (Sigma-Aldrich #A5040-25G) was used to anaesthetise dechorionated medaka embryos (20 mg/ml—20× stock solution diluted in 1xERM). Anaesthetised embryos were then mounted using low melting agarose (0.6–1%) (Biozyme Plaque Agarose #840101). Imaging was done in eight-well glass-bottomed dishes (Lab-Tek Chambered #1 Borosilicate Coverglass System 155411, T.S).

### Tail explants

Dechorionated Cab and Kaga embryos at the 15–16 somite stage were placed in Gibco $CO_2$ independent medium (ThermoFisher #18045054). Tails were cut 4–5 somites directly above the presomitic mesoderm (PSM). Dissected tails were then placed in individual wells of eight-well glass-bottomed dishes (Lab-Tek Chambered #1 Borosilicate Coverglass System 155411, Thermo Scientific Nunc, USA) with 200 µl of Gibco $CO_2$ independent medium (ThermoFisher #18045054).

### Hybridization chain reaction (HCR)

Cab and Kaga embryos at the 10 somite stage were fixed in 4% PFA in PtW for 12-24 h at 4 °C. Embryos were then washed 3× in PtW followed by dehydration and storage in MeOH at −20 °C. Medaka *mespb*, *pcdh10b* and *dll1* probes for hybridization chain reaction (Choi et al, 2018) were ordered from Molecular instruments (MI) and the protocol was carried out according to the manufacturer's guidelines. HCR amplifiers used were B1-546, B2-546, B2-647. Hoechst 33342 (Thermo Fischer #H3570) was used with a dilution of 1:500 of 10 mg/ml stock solution as a nuclear label.

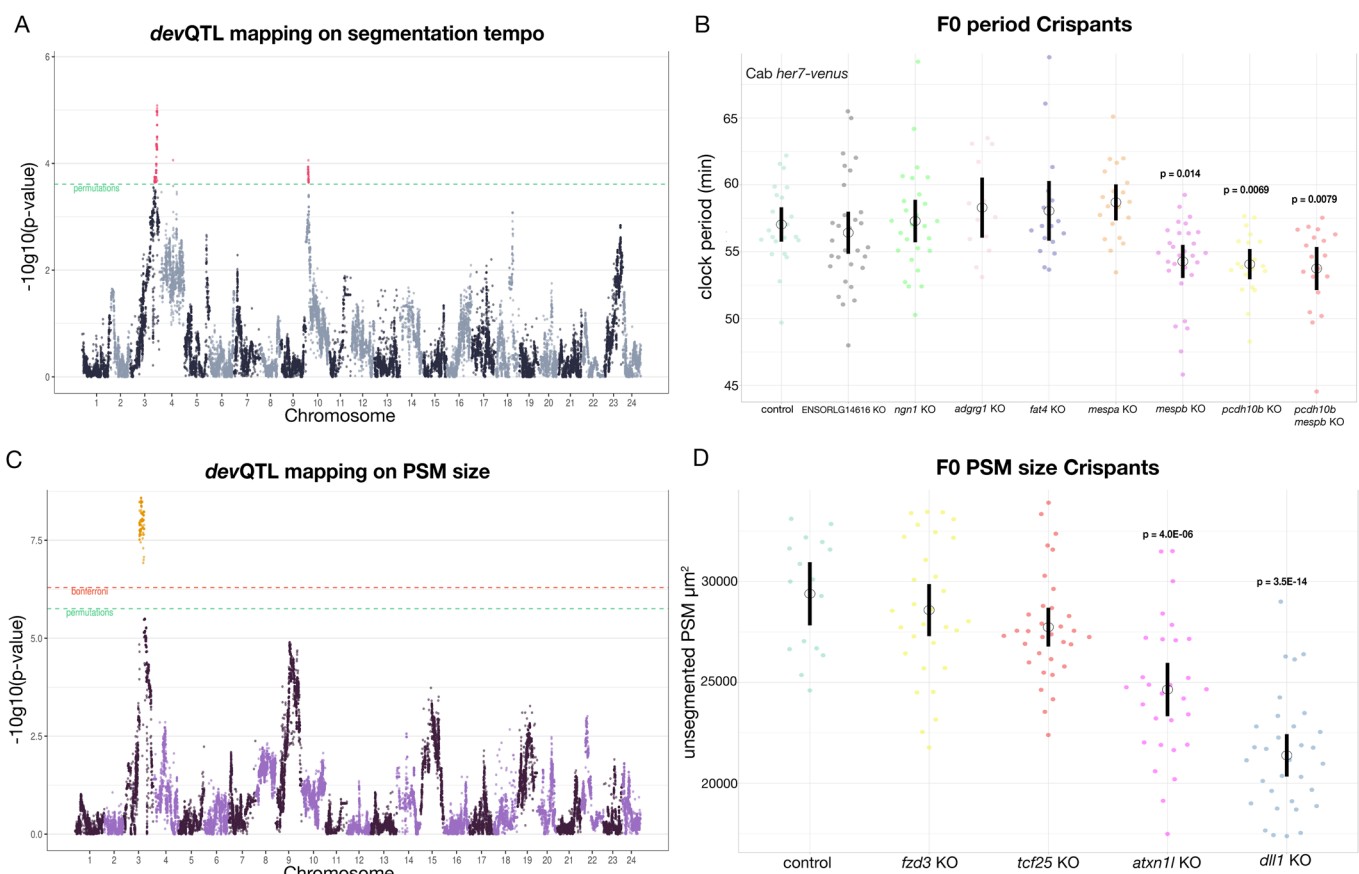

**Figure 4. *dev*QTL mapping and functional validation of segmentation timing and PSM size candidates.**

(A) *dev*QTL on segmentation timing performed on the F2 Kaga/Cab cross shows loci that passed the significance threshold located on chromosomes 3, 4 and 10. Manhattan plot of the genetic linkage results for the inverse-normalised clock period phenotype. Pseudo-SNPs with *P* values lower than the permutation significance threshold are highlighted in red. GCTA-LOCO mixed linear model was used, significance threshold was set at 10 permutations (red) (for details see "Methods") Permutation testing. (B) Endogenous *her7-venus* clock period measurements in Control, *ENSORLG14616, ngn1, adgrg1, fat4, mespa, mespb, pcdh10b, mespb+pcdh10b* F0 Cab Crispants imaged at the 10-11SS. Kruskal–Wallis' test *P* = 8.88E-08. Post-Hoc Dunn's test: only *mespb* (*P* adjusted=1.4E-02), *pcdh10b* (*P* adjusted=6.9E-03), *mespb +pcdh10b* (*P* adjusted=7.9E-03) show a significant difference in clock period as compared to control injected embryos. *N* = 23 control injected Cab *her7-venus* embryos, *N* = 30 *ENSORLG14616*, *N* = 27 *ngn1*, *N* = 13 *adgrg1*, *N* = 17 *fat4*, *N* = 20 *mespa*, *N* = 29 *mespb*, *N* = 20 *pcdh10b*, *N* = 19 *pcdh10b+mespb* CRISPR/Cas9 injected into Cab *her7-venus*. Black bars indicate 95% confidence interval, Black circle indicates the mean. (C) *dev*QTL on unsegmented PSM size performed on the F2 Kaga/Cab cross shows a single locus that passed the significance threshold located on chromosome 3 at a distinct genomic coordinate from that found on chromosome 3 for the segmentation timing *dev*QTL hits. Manhattan plot of the genetic linkage results for the unsegmented PSM size phenotype. Pseudo-SNPs with *P* values lower than the permutation (green) and Bonferroni (red) significance thresholds are highlighted in orange. GCTA-LOCO mixed linear model was used, significance threshold was set at 10 permutations and Bonferroni *P* value was set by dividing 0.05 by the number of pseudo-SNPs in the model (for details see "Methods"). (D) Unsegmented PSM size Cab F0 CRISPR/Cas9 knock-outs performed on candidate genes from the *dev*QTL mapping results in (C) on Control, *fzd3, tcf25, atxn1l* and *dll1* F0 Cab Crispants imaged at the 10-11SS. Both *atxn1l* and *dll1* Crispants showed significantly smaller unsegmented PSM size (24,645 μm² (SD +/− 3426)) and (21,382 μm² (SD +/− 2868)) respectively than Cab control Cas9 mRNA injected embryos (29,392 μm² (SD +/− 2851)). One-way ANOVA test *P* = 7.5E-15. Post-Hoc Dunnett's test: only *atxn1l* (*P* = 4.0E-06), *dll1* (*P* = 3.5E-14) show a significant difference in PSM size as compared to control injected embryos. *N* = 16 control injected embryos, *N* = 30 *fzd3* Crispants *N* = 33 *tcf25* Crispants *N* = 29 *atxn1l* Crispants *N* = 32 *dll1* Crispants. Black bars indicate 95% confidence interval, Black circle indicates the mean. Source data are available online for this figure.

## Microscopy

All embryo screening was done on a Nikon SMZ18 fluorescence stereoscope equipped with a camera. ALC and HCR image acquisition was done either on Nikon SMZ18 fluorescence stereoscope equipped with a camera or on a laser-scanning confocal Leica SP8 (CSU, White Laser) microscope, 20× and 40× objectives were used during image acquisition depending on the experimental sample. For the SP8 confocal equipped with a white laser, the laser emission was matched to the spectral properties of the fluorescent protein of interest. For all F0, F1, F2 imaging of

embryos and for all CRISPR/Cas9 knock-outs (KOs) image acquisition was performed using two Zeiss LSM780 laser-scanning confocal microscopes with a temperature control box and an Argon laser at 488 nm, imaged through a 20× plan apo objective (numerical aperture 0.8). All embryos were imaged at 10-11SS unless otherwise stated. Temperature on the incubator box of both microscopes was set at 30 °C. To account for slight differences in actual temperature in the wells between the two microscopes used for the F2 data acquisition: mean and intercept period measurements were normalised and plotted by microscope, normalisation was done either by using inverse normalisation with

the following formula:

$$z_{i,j} = qnorm(rank(y_{i,j})/(N_j + 0.5))$$

Where $rank$ $(y_{i,j})$ is the sample rank of observation $i$ within microscope $j$, $N_j$ is the sample size for microscope $j$, and $qnorm$ calculates the percentile value based on the normal distribution (Wichura, 1988) or by equating the mean period of all samples on one microscope (reference) to the mean period of all samples on the other microscope. The difference between the mean measurements on the two microscopes translates to 3.5 min for mean period and 4.0 min for intercept period. For all other experiments either one microscope (reference) was used or a temperature sensor probe GMH 3200 series Thermocouple (Greisinger) was placed into the imaging wells and temperature was measured throughout imaging to ensure equivalent temperatures were measured between the two microscopes.

## Data analysis

Open-source ImageJ/Fiji software (Schindelin et al, 2012) was used for analysis and editing of all images post image acquisition. Stitching was performed using 2D and 3D stitching plug-ins on ImageJ/Fiji. For extracting quantitative values of posterior period oscillations in the endogenous her7-Venus line, the time-series movie was first Gaussian blurred (sigma 8) in ImageJ/Fiji then ROI manager was used to define fluorescence intensity within a circle (area 300–600 $\mu m^2$), at the posterior most tip of the tail, the circle was manually moved to track the movement and growth of the tails over the course of imaging, fluorescent intensity measurements were then concatenated for every time point and extracted from the time-series using a custom made Fiji macro-script provided in Data EV1. For F0 segmentation time estimation in the Cab, Kaga, HNI, Ho5 and HdrII strains we used segment boundary formation obtained from bright-field live-imaging to determine segment forming time. The time it took to form 5 consecutive segments was calculated for each embryo to get an estimation of segmentation time. For presomitic mesoderm (PSM) size area measurements were done using polygon selection in Fiji on brightfield tail images of 10-11SS embryos on the unsegmented tissue at the tip of the tail. Presomitic mesoderm (PSM) length measurements were done using the segmented line tool in Fiji on brightfield tail images of 10-11SS embryos on the unsegmented tissue at the tip of the tail. Somite length and area measurements were done on the first pair of nascent somites using the line or polygon tool in Fiji on brightfield tail time-lapse imaging of 10-11SS embryos. For volumetric egg measurements the eggs were approximated as oblate spheroids (Iwamatsu, 2004) and the following equation was used V = 4/3.π.(B)². c. where V= volume b= semi-major axis and c= semi-minor axis. Data was plotted using *ggplot2* and *gganimate* in R software or using *PlotTwist* (Goedhart, 2020) and *PlotsofData* (Postma and Goedhart, 2019). PGLS analysis was performed in R using *caper* (Orme, 2013), we relied on molecular phylogenies of *Oryzias* fish previously reported (Takehana et al, 2005). Pearson's product moment correlations, F-test for equality of variances, Welch two sample *t* tests, Kruskal–Wallis test, one-way ANOVA test, Dunnet's test, Tukey HSD test and Dunn's test were all calculated and plotted in R version 4.2.2. No blinding was performed. All replicates are biological replicates unless otherwise indicated. No statistical

method was used to predetermine group size. Replicates were estimated based on preliminary data.

## Wavelet analysis and period extraction

Raw fluorescent intensity measurements were used for wavelet analysis and period extraction using PyBoat (Mönke et al, 2020). The following settings were used for all samples: sampling interval 10 min, cut-off period was set at 100 min, window size was set at 150 min, the smallest period was set at 40 min, the number of periods to scan was set at 200, the highest period was set at 100 mins, detrended signal and normalisation with envelope was used on all samples. The continuous maximum ridge connecting the wavelet power was then plotted. Data within the COI (cone of influence) were then extracted to get period, phase, amplitude and power for each analysed sample. Period values, one for each 10 min sampling interval, for a total of 300 min were then used for all subsequent analysis, either to generate mean period plots (average period in 300 min interval) or intercept period plots (y-intercept of fitted line on 300 min interval period measurements). The clock period shown in the main figures corresponds to intercept period measurements unless otherwise stated. Both mean and intercept period measurements are shown in the Supplementary Figures.

## Bulk RNA-sequencing

Dechorionated Cab, Kaga and *mespb* mutant embryos at the 13-14 somite stage were placed in Gibco $CO_2$ independent medium (ThermoFisher #18045054). Using a forceps and a scalpel, tails were cut directly at the unsegmented presomitic mesoderm (PSM) border. Five dissected tails in three replicates for Cab and Kaga and another five dissected tails in three replicates for Cab and *mespb* mutant embryos were then disrupted and homogenised for total RNA extraction using RNeasy Plus Micro Kit (Qiagen #74034) following the manufacturer's guidelines. The integrity and concentration of the extracted RNA was checked by using Agilent Bioanalyzer with the RNA 6000 Nano Assay kit. Libraries were prepared using the NEBNext Ultra II Directional RNA Library Prep Kit for Illumina (New England Biolabs) together with the NEBNext Poly(A) mRNA Magnetic Isolation Module (New England Biolabs) using the long inserts version of the manufacturer's protocol. Briefly, these modifications consisted of 7 min of fragmentation, 50 min extension for the first strand synthesis, 1:100 dilution of the adaptor and size selection for a 400 base pair long insert. The libraries were quantified using the Qubit HS DNA assay as per the manufacturer's protocol. For the measurement, 1 µL of sample in 199 µL of Qubit working solution was used. The quality and molarity of the libraries was assessed using Agilent Bioanalyzer with the DNA HS Assay kit as per the manufacturer's protocol. The assessed molarity was used to equimolar combine the individual libraries into one pool for sequencing. The pool was sequenced on Illumina NextSeq2000 (Illumina, San Diego, CA, USA) using a P3 flowcell and reading 2 ×150 bases. Sequencing files were demultiplexed using FastQC (version 0.11.9) and the output was collated using MultiQC (version1.10) (Ewels et al, 2016). Sequencing reads were aligned using STAR (version 2.7.9a) (Dobin et al, 2013) to the medaka genome (version ASM223467v1) with default parameters. The gene count tables

were computed during the alignment with STAR on the medaka gene model (version ASM223467v1.103). Differential analysis was performed using DESeq2 in R. All data was deposited on public repository Biostudies Acc. Number: E-MATB-13927 and E-MATB-13928.

## CRISPR/Cas9 knock-outs

Embryos (WT Cab or *her7-venus* Cab) were injected at the 1 cell stage. Cas9 mRNA was obtained from pCS2-Cas9 (Addgene #47322) as previously described (Gagnon et al, 2014; Seleit et al, 2021) In-vitro transcription was carried out using mMachine SP6 Transcription Kit (Invitrogen #AM1340) following the manufacturer's guidelines. RNA cleanup was carried out using RNAeasy Mini Kit (Qiagen #74104). 2-3 synthetic gRNAs per gene targeting exonic regions were designed using CCTop (Stemmer et al, 2015) and ordered from Sigma-Aldrich (spyCas9 sgRNA, 3 nmol, HPLC purification, no modification). A list of all gRNAs and their corresponding target genes used in this study is provided in Data EV6. The injection mix consists of 15 ng/µL for each gRNA and 75 ng/µL Cas9 mRNA. For control injections only the Cas9 mRNA was injected.

## DNA extraction and library preparation for F0 Cab and Kaga and F1 Kaga/Cab hybrid samples

DNA from two separate stage 42 embryos of F0 Cab and F0 Kaga and one stage 42 embryo of F1 hybrid Kaga/Cab was extracted using DNeasy Blood and Tissue Kit (Qiagen #69504) following the manufacturer's guidelines. The extracted DNA was quantified using the Invitrogen Qubit Flex, with the Qbit dsDNA High Sensitivity assay as per the manufacturer's protocol. For the measurement, 1 µL of sample in 199 µL of Qubit working solution was used. Samples were then standardised and 1 µg of material was used as an input for library preparation of both F0 Cab and F0 Kaga, while the input amount was 750 ng for the F1 Kaga/Cab hybrid sample. Libraries were prepared using the NEBNext UltraII DNA Library Prep Kit (New England Bioalbs) with NEBNext Multiplex Oligos for Illumina (Unique Dual Index UMI Adaptors DNA Set 1) according to the manufacturer's instructions without PCR. A size selection for 500 base pairs insert size was performed. The libraries were quantified using the Qubit HS DNA assay as per the manufacturer's protocol. For the measurement, 1 µL of sample in 199 µL of Qubit working solution was used. The quality and molarity of the libraries was assessed using Agilent Bioanalyzer with the DNA HS Assay kit as per the manufacturer's protocol. The assessed molarity was used to equimolar combine the individual libraries into one pool for sequencing for the F0 Cab and F0 Kaga samples. The pool (F0 samples) and the hybrid F1 sample were both sequenced on Illumina NextSeq500 (Illumina, San Diego, CA, USA) using a MID output kit and reading 2 ×150 bases. The sequencing data for this study have been deposited in the European Nucleotide Archive (ENA) at EMBL-EBI under accession number PRJEB59222 (https://www.ebi.ac.uk/ena/browser/view/PRJEB59222).

## DNA extraction and library preparation for F2 samples

After imaging, F2 embryos were recovered and grown in individual wells of 12-well plates (Thermo-Fisher #150628) until stage 41–42. Samples were then placed in 1.5 ml Eppendorf tubes, snap-frozen

and stored in −80 °C. For DNA extraction of all F2 samples, the following protocol was used: 40 µL Fin clip Buffer was added to snap-frozen embryos. Embryos were then Incubated at 65 °C overnight. In total, 80 µL distilled $H_2O$ was then added and samples were incubated for 10 min at 95 °C. Samples were then centrifuged at 10,000 rpm at 4 °C for 25 min. Supernatant containing genomic DNA was then used for subsequent library preparation. Fin clip buffer is composed of 100 ml 2 M Tris pH 8.0, 5 ml 0.5 M EDTA pH 8.0, 15 ml 5 M NaCl, 2.5 ml 20% SDS, $H_2O$ to 500 ml, sterile filtered. The extracted DNA per sample was quantified using the Invitrogen Qubit Flex, with the Qbit dsDNA High Sensitivity assay as per the manufacturer's protocol. For the measurement, 1 µL of sample in 199 µL of Qubit working solution was used. Each sample was then standardised with water to a final concentration of 2 ng/µL. Sequencing libraries were prepared as previously described (Hennig et al, 2018; Picelli et al, 2014) using the automated liquid handler Biomek i7 system (Beckman Coulter). In short, 1.25 µL of each sample was taken into a tagmentation reaction containing 1.25 µL of Dimethylformamide, 1.25 µL of tagmentation buffer (40 mM Tris-HCl pH 7.5, 40 mM $MgCl_2$) and 1.25 µL of an in-house generated and purified Tn5 (Hennig et al, 2018) and diluted to a 1:100 ratio with water. The mixture was incubated at 55 °C for 3 min. After that, 1.25 µL of SDS 0.2% was added to stop the tagmentation reaction followed by a 5 min incubation at room temperature. The resulting fragments were then amplified by PCR using 6.75 µL of KAPA 2 × HiFi master mix, 0.75 µL of Dimethyl sulfoxide and 2.5 µL of dual indexed primers. Cycling conditions were as follows: 3 min at 72 °C; 30 s at 95 °C; 12 cycles of 20 s at 98 °C, 15 s at 58 °C, 30 s at 72 °C; 3 min at 72 °C. The resulting PCR products were then pooled together by combining 4 µL of each desired sample. Finally, the pools were size selected using two rounds of magnetic SPRI bead purification (0.6×) and quantified using the Qbit dsDNA High Sensitivity assay. Samples were sequenced either on Illumina Hiseq4000 (150 paired end) (Illumina, San Diego, CA, USA) or Illumina Nextseq2000 (150 paired end) (Illumina, San Diego, CA, USA). All sequencing data is deposited at the ENA under study number: PRJEB59222.

## Whole genome sequencing, alignment, and variant calling for Cab and Kaga F0 and Kaga/Cab hybrid F1s

Coverage for each sample was measured using SAMtools (Danecek et al, 2021) with a mean of ~26× for F0 Cab, ~29× for F0 Kaga and ~59× for Kaga/Cab hybrid. Reads were then aligned to the medaka HdrR reference (Ensembl release 104, build ASM223467v1) using *BWA-MEM2* (Vasimuddin et al, 2019), sorted the aligned.sam files, marked duplicate reads, merged the paired reads with the *Picard* toolkit, (Picard Toolkit, 2019) and indexed the.bam files with *SAMtools* (Li et al, 2009). The Snakemake pipeline used to map and align these samples can be found in the GitHub repository here: https://github.com/birneylab/somites. To call variants, we followed the *GATK* best practices (to the extent they were applicable) (DePristo et al, 2011; McKenna et al, 2010) with *GATK*'s HaplotypeCaller and GenotypeGVCFs tools (Poplin et al, 2018), then merged all calls into a single.vcf file with *Picard*. Finally, we extracted the biallelic calls for Cab and Kaga with bcftools (Danecek et al, 2021), counted the number of SNPs within non-overlapping, 5-kb bins, and calculated the proportion of SNPs within each bin that were homozygous to generate Fig. 3H,I and Fig. S3 using R

version 4.2.2 (R Core Team, 2021), the *tidyverse* suite of R packages (Wickham et al, 2019) and *circlize* (Gu et al, 2014). To assess whether the low homozygosity observed in the Kaga strain was caused by a reference bias, we also aligned the reads of the Kaga F0 sample to the northern Japanese HNI reference (Ensembl release 105 build ASM223471v1) to generate Fig. S3C using the same process as above.

## Whole genome sequencing and alignment on Kaga/Cab F2s

For all F2 samples the sequencing reads were aligned to the HdrR reference (Ensembl release 104, build ASM223467v1) using *BWA-MEM2* (Vasimuddin et al, 2019), sorted the aligned.sam files, marked duplicate reads, merged the paired reads with the *Picard* toolkit and indexed the.bam files with *SAMtools* (Li et al, 2009) in the same manner as for the F0 and F1 samples. To map the aligned F2 sequences to the genomes of their parental strains, we selected only biallelic SNPs that were homozygous-divergent in the F0 generation (that is, homozygous for the reference allele in Cab and homozygous for the alternative allele in Kaga, or vice versa) *and* heterozygous in the F1 generation. The number of SNPs that met these criteria per chromosome are set out in Fig. S7. To call variants in the shallow-sequenced F2 samples, we applied a method involving the use of a Hidden Markov Model (HMM) to classify regions of the genome as one of the three genotypes (AA, AB, or BB), based on the proportion of reads that supported either parental allele, as described previously (Rowan et al, 2015). We accordingly used *bam-readcount* (Khanna et al, 2022) to count the number of reads that supported either the Cab or the Kaga allele for all SNPs that met the criteria described above (i.e. biallelic SNPs that were homozygous-divergent in the parental Cab and Kaga F0 samples), summed the read counts within 5 kb blocks across the genome, and calculated the frequency of reads within each bin that supported the Kaga allele. This generated a value for each bin between 0 and 1, where 0 signified that all reads within that bin supported the Cab allele, and 1 signified that all reads within that bin supported the Kaga allele. Bins containing no reads were assigned a value of 0.5. We then used these values for all F2 individuals as the input to an HMM with the software package *hmmlearn* (Hmmlearn/Hmmlearn, 2022), which we applied to classify each bin as one of three states, with state 0 corresponding to homozygous-Cab, state 1 corresponding to heterozygous, and state 2 corresponding to homozygous-Kaga. Across each chromosome of every sample, the output of the HMM was expected to produce a sequence of states. Based on previous biological knowledge that crossover events occur on average less than once per chromosome (Haenel et al, 2018) we expected to observe the same state persisting for long stretches of the chromosome, only changing to another state between 0 and 3 times, and rarely more. To achieve this, we adjusted the HMM's transition probabilities to be nearly 0, and the Gaussian emission probabilities for each state to have a variance of 0.8, which resulted in long "blocks" of the same genotype call across the chromosome with only a small number of average transitions (i.e. crossover events) per chromosome. We then used the R package *karyoploteR* (Gel and Serra, 2017) to generate the recombination block plot shown in Fig. 3J. Figure S3 shows the proportion of 5-kb bins called as either homozygous-Cab, heterozygous, or homozygous-Kaga within each F2 sample

(points). The ordinary expectation for the ratios would be 0.25, 0.5, and 0.25, respectively. However, we observe a skew towards homozygous-Cab and away from homozygous Kaga. This was likely caused by the lower level of homozygosity in Kaga, and also potentially a degree of hybrid incompatibility between Cab and Kaga, given the two strains were derived from populations that are at or beyond the point of speciation. In the downstream analysis, we excluded the 22 samples that showed poor coverage across the genome, leaving $N = 600$ for the genetic association testing. For these remaining samples, we "filled" the bins with missing genotypes based on the call of the previous called bin, or if unavailable (e.g. the missing bin was at the start of the chromosome), then the next called bin. We used these filled genotype calls for the genetic association tests described below.

### *dev*QTL and VEP output

We used the recombination blocks called by the HMM as pseudo-SNPs in an F2-cross *dev*QTL. To detect associations between the pseudo-SNPs and the two phenotypes of interest, we used a linear mixed model (LMM) as implemented in GCTA (Yang et al, 2011) For the genetic relationship matrix (GRM), we additionally used the leave-one-chromosome-out implementation of GCTA's LMM, which excludes the chromosome on which the candidate SNP is located when calculating the GRM. A GRM constructed from the entire genome is presented as a heatmap in Fig. S3E with each sample represented on each axis, and lighter colours representing a higher degree of relatedness between a pair of samples. The square in the top right-hand corner is created by samples ~550-648, which have distinct genotypes to the rest of the samples due to their having been bred from different F1 parents. To set the significance threshold, we permuted the phenotype across samples using 10 different random seeds, together with all covariates when included, and ran a separate linkage model for each permutation. We then set the lowest $P$ value from all 10 permutations as the significance threshold for the non-permuted model. We additionally applied a Bonferroni correction to our $P$ values by dividing $\alpha$ (0.05) by the number of pseudo-SNPs in the model, and set this as a secondary threshold. For the pseudo-SNPs (5-kb regions) which returned a lower $P$ value than the significance level set by the permutations, we ran Ensembl's Variant Effect Predictor to identify the variants that are predicted to be more likely to disrupt the functions of the sequence.

## Data availability

All data are available within the article, supplementary files and source data files. All materials used in this study are available by request from the corresponding author. Source data are provided with this paper. The sequencing data for this study have been deposited in the public repository of the European Nucleotide Archive (ENA) at EMBL-EBI under accession number PRJEB59222. All code used in this study is available at https://github.com/birneylab/somites.

The source data of this paper are collected in the following database record: biostudies:S-SCDT-10_1038-S44318-024-00186-2.

## Peer review information

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

## Acknowledgements

We would like to thank all members of the Aulehla lab for discussions throughout the project and Jordi Van Gestel for comments on the manuscript. The European Molecular Biology Laboratory (EMBL-Heidelberg) Genecore is acknowledged for support in WGS data acquisition and analysis. We would like to thank Vladimir Benes and members of his team at Genecore EMBL Heidelberg for continuous help and support, Tobias Rausch for computational work on the WGS data and Mireia Osuna Lopez for help in library preparation of all WGS DNA and RNA-seq data and Jonathan Landry for help with the RNA-seq data analysis. The EMBL-EBI's high performance computing cluster is acknowledged for providing the computational resources required for the *dev*QTL analysis. We would like to thank Sabine Goergens and her team at EMBL Laboratory Animal Resource (LAR) Team fish facility for their excellent, daily animal husbandry and expert support. We would like to thank the National Bioresource Project (NBRP) Japan for access to medaka hatching enzyme and for *O. hubbsi, O. mekongensis, O. minutillus* fish. This work was supported by The European Molecular Biology Laboratory (EMBL).The EMBL interdisciplinary Postdoc (EIPOD4) under Marie Sklodowska-Curie Actions Cofund (grant agreement no. 847543) fellowship for funding to A.S. The EMBL International PhD Programme fellowship to IB. The European Research Council under an ERC consolidator grant agreement no. 866537 to AA.

## Author contributions

**Ali Seleit**: Conceptualization; Data curation; Formal analysis; Validation; Investigation; Visualization; Methodology; Writing—original draft; Writing—review and editing. **Ian Brettell**: Data curation; Software; Formal analysis; Investigation; Visualization; Methodology; Writing—review and editing. **Tomas Fitzgerald**: Data curation; Software; Formal analysis; Methodology. **Carina Vibe**: Validation; Methodology; Writing—review and editing. **Felix Loosli**: Conceptualization; Resources; Writing—review and editing. **Joachim Wittbrodt**: Conceptualization; Resources; Writing—review and editing. **Kiyoshi Naruse**: Conceptualization; Resources. **Ewan Birney**: Conceptualization; Resources; Supervision; Funding acquisition; Methodology; Project administration; Writing—review and editing. **Alexander Aulehla**: Conceptualization; Resources; Supervision; Funding acquisition; Methodology; Writing—original draft; Project administration; Writing—review and editing.

Source data underlying figure panels in this paper may have individual authorship assigned. Where available, figure panel/source data authorship is listed in the following database record: biostudies:S-SCDT-10_1038-S44318-024-00186-2.

## Funding

## Disclosure and competing interests statement

The authors declare no competing interests.

# Expanded View Figures

**Figure EV1.  Scaling of embryonic segmentation timing and size in the *Oryzias* genus.**

(**A**) Pearson's correlation of average trait values for adult mass and adult length (11 months) in *O. minutillus* (pink), *O. hubbsi* (blue), *O. mekongensis* (orange), *O. sakaizumii* (light blue) and *O. latipes* (magenta) shows a positive correlation $R = 0.99$ $P$ value = 1.0E-03. Black dotted line= linear fit on average trait values, vertical and horizontal lines on both axes represent standard deviation (SD). $N = 10$ *O. minutillus*, $N = 10$ *O. minutillus*, $N = 12$ *O. hubbsi*, $N = 10$ *O. mekongensis*, $N = 13$ *O. sakaizumii*, $N = 15$ *O. latipes* for adult length and $N = 6$ each *O. hubbsi, O. mekongensis, O. sakaizumii, O. latipes* for adult mass. (**B**) Pearson's correlation of average trait values for adult mass and embryonic segmentation rate in *O. minutillus* (pink), *O. hubbsi* (blue), *O. mekongensis* (orange), *O. sakaizumii* (light blue) and *O. latipes* (magenta) shows a positive correlation $R = 0.94$ $P$ value = 1.9E-02. Black dotted line= linear fit on average trait values, vertical and horizontal lines on both axes represent standard deviation (SD). $N = 10$ *O. minutillus*, $N = 10$ *O. minutillus*, $N = 12$ *O. hubbsi*, $N = 10$ *O. mekongensis*, $N = 13$ *O. sakaizumii*, $N = 15$ *O. latipes* for adult length and $N = 11$ *O. minutillus*, $N = 10$ *O. hubbsi*, $N = 10$ *O. mekongensis*, $N = 20$ *O. sakaizumii*, $N = 30$ *O. latipes* for segmentation rate. (**C**) Bright-field image of unsegmented presomitic mesoderm (PSM) and nascent somites at the 10-11 somite stage (SS) Cab F0 medaka embryo. Yellow dotted line delineates the unsegmented PSM area while white dotted lines delineate the area of nascent somites. Scale bar= 50 μm. (**D**) Schematic map of mainland Japan (showing Honshu, Shikoku and parts of Kyushu) highlighting the sites of the original medaka populations. The northern *Oryzias sakaizumii*: HNI (blue) and Kaga (red) diverged from the southern *Oryzias latipes*: Cab (turquoise), HdrII (magenta) and Ho5 (yellow) 18 million years ago. (**E**) Pearson's correlation of average trait values for nascent somite size and somite length measured at 10-11 somite stage in *O. minutillus* (pink), *O. hubbsi* (blue), *O. mekongensis* (orange), *O. sakaizumii* Kaga (red), HNI (light blue) and *O. latipes* Cab (green), HdrII (magenta), Ho5 (yellow) embryos shows a positive correlation $R = 0.95$ $P$ value = 1.5E-02. Black dotted line= linear fit on average trait values, vertical and horizontal lines on both axes represent standard deviation (SD). Individual data points are shown for each population $N = 11$ *O. minutillus*, $N = 13$ *O. hubbsi*, $N = 18$ *O. mekongensis*, $N = 19$ Cab, $N = 10$ HdrII, $N = 7$ Ho5, $N = 20$ Kaga, $N = 10$ HNI. (**F**) Pearson's correlation of average trait values for unsegmented PSM length and unsegmented PSM area measured at 10-11 somite stage in *O. minutillus* (pink), *O. hubbsi* (blue), *O. mekongensis* (orange), *O. sakaizumii* Kaga (red), HNI (light blue) and *O. latipes* Cab (green), HdrII (magenta), Ho5 (yellow) embryos shows a positive correlation $R = 0.99$ $P$ value = 3.2E-04. Black dotted line= linear fit on average trait values, vertical and horizontal lines on both axes represent standard deviation (SD). Individual data points are shown for each population $N = 11$ *O. minutillus*, $N = 13$ *O.hubbsi*, $N = 18$ *O. mekongensis*, $N = 19$ Cab, $N = 10$ HdrII, $N = 7$ Ho5, $N = 20$ Kaga, $N = 10$ HNI. (**G**) Pearson's correlation of average trait values for unsegmented PSM length and somite length measured at 10-11 somite stage in *O. minutillus* (pink), *O. hubbsi* (blue), *O. mekongensis* (orange), *O. sakaizumii* Kaga (red), HNI (light blue) and *O. latipes* Cab (green), HdrII (magenta), Ho5 (yellow) embryos shows a positive correlation $R = 0.96$ $P$ value = 9.1E-03. Black dotted line= linear fit on average trait values, vertical and horizontal lines on both axes represent standard deviation (SD). Individual data points are shown for each population $N = 11$ *O. minutillus*, $N = 13$ *O. hubbsi*, $N = 18$ *O. mekongensis*, $N = 19$ Cab, $N = 10$ HdrII, $N = 7$ Ho5, $N = 20$ Kaga, $N = 10$ HNI. (**H**) Axis segmentation rate from brightfield time-lapse imaging of Kaga and Cab F0 embryos measured at the 10-11 somite stage calculated from the time it takes to form 5 consecutive pairs of somites. Kaga embryos have a faster segmentation period (52.88 min (SD +/− 2.42)) than Cab (60.84 min (SD +/− 2.52)). Welch two sample $t$ test $P = 4.3\text{E-}12$. Black circle = mean Black line = 95% confidence interval. $N = 20$ Kaga embryos, $N = 19$ Cab embryos. (**I**) Ex-vivo explant axis segmentation rate in bright-field imaging of Kaga and Cab F0 embryos at the 15-16 somite stage calculated from the time it takes to form 5-6 consecutive pairs of somites. Kaga embryos have a faster ex-vivo segmentation period (63.47 min (SD +/− 4.2)) than Cab (73.75 min (SD +/− 5.9)). Welch two sample $t$ test $P = 2.8\text{E-}05$. Black circle = mean Black line = 95% confidence interval. $N = 13$ Kaga embryos, $N = 14$ Cab embryos.

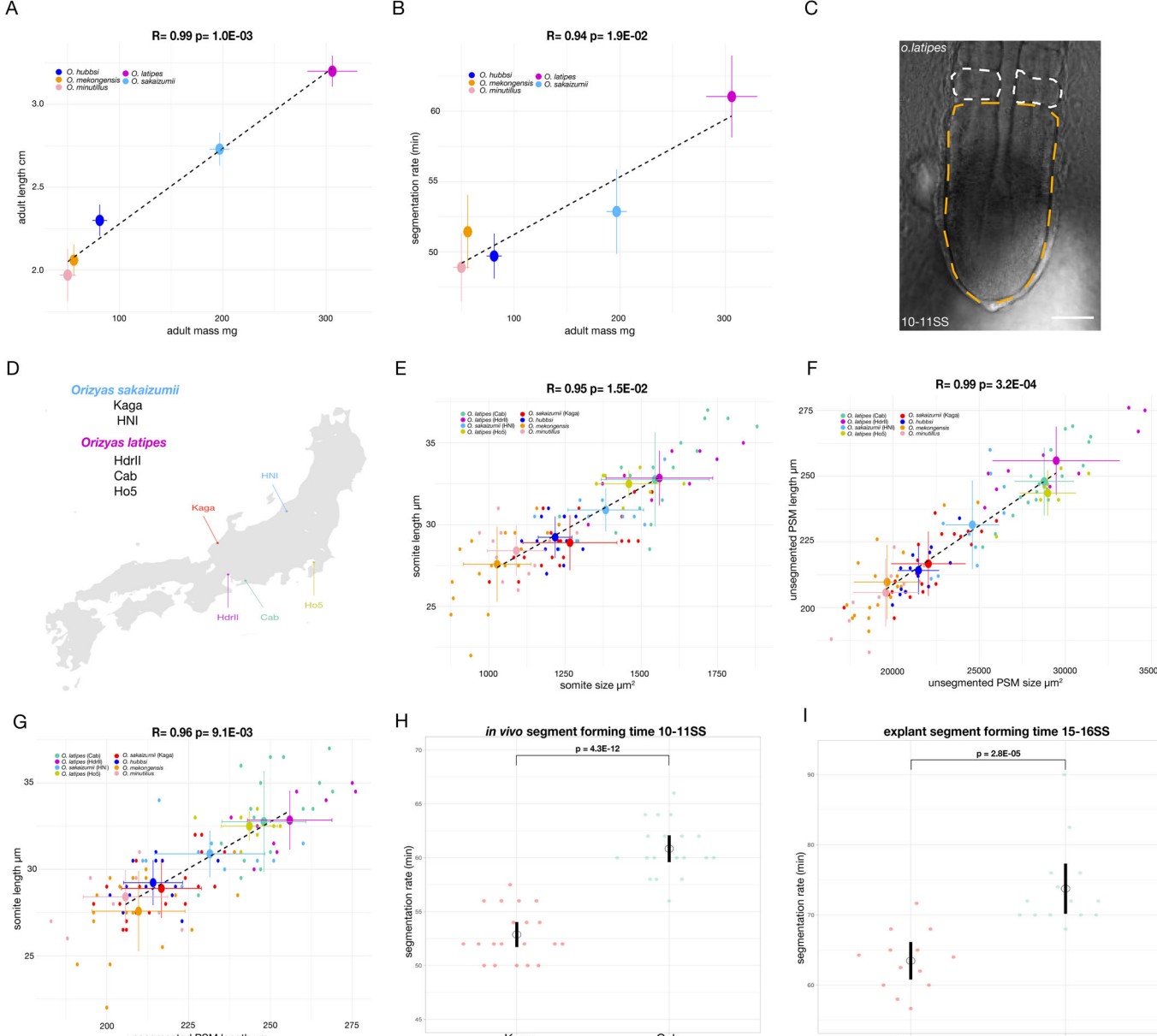

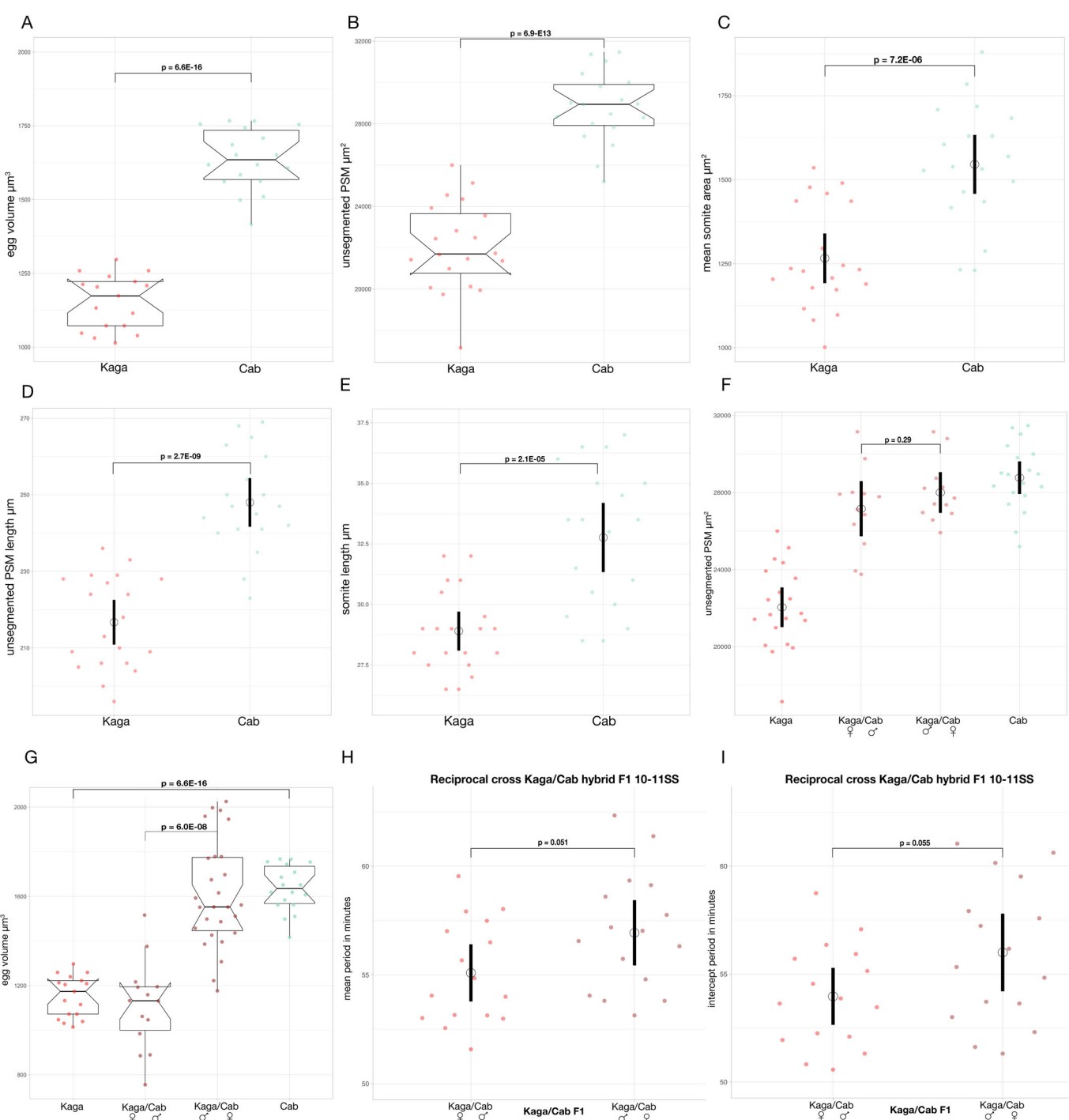

◀

**Figure EV2. Kaga and Cab F0 and reciprocal hybrid F1 segmentation timing and size measurements.**

(A) F0 Kaga have smaller fertilised egg volume (1152 µm³ (SD +/− 93,0)) than Cab (1636 µm³ (SD +/− 103)). Welch two sample $t$ test $P = 6.6E$-16. $N = 17$ Kaga $N = 18$ Cab eggs. (B) Area of unsegmented PSM at the 10-11 somite stage of Kaga and Cab F0 embryos. Kaga embryos have smaller unsegmented PSM (22,048 µm² (SD +/− 2145)) compared to Cab (28,770 µm² (SD +/− 1709)). Welch two sample $t$ test $P = 6.9E$-13. $N = 20$ Kaga $N = 19$ Cab embryos. (C) somite area of nascent pair of somites in Kaga and Cab F0 embryos at the 10-11 somite stage. Kaga embryos have smaller somites (1265 µm² (SD +/− 154)) than Cab (1545 µm² (SD +/− 177)) embryos. Black circle = mean Black line = 95% confidence interval. Welch two sample $t$ test $P = 7.20E$-06. $N = 20$ Kaga embryos, $N = 19$ Cab embryos. (D) Length of unsegmented PSM at the 10SS of Kaga and Cab F0 embryos. Kaga embryos have shorter unsegmented PSM (216.7 µm (SD +/− 12.19)) compared to Cab (248 µm (SD +/− 12.83)). Black circle = mean Black line = 95% confidence interval. Welch two sample $t$ test $P = 2.7E$-09. $N = 20$ Kaga, $N = 19$ Cab embryos. (E) Length of nascent somite measured at the 10SS of Kaga and Cab F0 embryos. Kaga embryos have shorter somites (28.9 µm (SD +/− 1.67)) compared to Cab (32.76µm (SD +/− 2.88)). Black circle = mean Black line = 95% confidence interval. Welch two sample $t$ test $P = 2.1E$-05. $N = 20$ Kaga $N = 19$ Cab embryos. (F) Area of unsegmented PSM µm² at the 10-11SS of Kaga/Cab F1 reciprocal cross compared to the paternal F0 Kaga and Cab populations. Kaga/Cab F1 embryos coming from Kaga females crossed to Cab male embryos have unsegmented PSM size of (2716 µm² (SD +/− 1592)) compared Kaga/Cab F1 embryos coming from Kaga males crossed to Cab females (28,004 µm² (SD +/− 2154)). While the Kaga F0 population embryos have an unsegmented PSM size of (22,048 µm² (SD +/− 2145)) compared to Cab (28,770 µm² (SD +/− 1709)). Black circle = mean Black line = 95% confidence interval. Welch two sample $t$ test $P = 0.29$. $N = 20$ Kaga, $N = 19$ Cab embryos. $N = 12$ Kaga/Cab F1 embryos coming from a cross of Kaga females to Cab males, $N = 12$ Kaga/Cab F1 embryos coming from a cross of Kaga males to Cab females. (G) Fertilised egg volume of Kaga/Cab F1 reciprocal cross shows a significant maternal effect. Kaga/Cab F1 embryos coming from Kaga females crossed to Cab male embryos have fertilised egg volume of (1110 µm³ (SD +/− 198)) compared Kaga/Cab F1 embryos coming from Kaga males crossed to Cab females 1605 µm³ (SD +/-236)). Welch two sample $t$ test $P = 6.0E0$-08. $N = 14$ Kaga/Cab F1 embryos coming from a cross of Kaga females to Cab males, $N = 27$ Kaga/Cab F1 embryos coming from a cross of Kaga males to Cab females. F0 Kagas have smaller fertilised egg volume (1152 µm³ (SD +/− 93.0)) than Cab (1636 µm³ (SD +/− 103)). Black circle = mean Black line = 95% confidence interval. Welch two sample $t$ test $P = 6.6E$-16. $N = 17$ Kaga, $N = 18$ Cab eggs. (H) endogenous *her7-venus* mean period measurements in hybrid F1 Kaga/Cab reciprocal cross at the 10-11SS shows non-significant difference. Kaga/Cab F1 embryos coming from Kaga females crossed to Cab males have a mean *her7-venus* period of (55.09 min (SD +/− 2.39)) while Kaga/Cab F1 embryos coming from Kaga males crossed to Cab females have a mean period of (56.94 min (SD +/− 2.73)). Black circle = mean Black line = 95% confidence interval. Welch two sample $t$ test $P = 0.051$. $N = 16$ Kaga/Cab F1 embryos coming from a cross of Kaga females to Cab males, $N = 16$ Kaga/Cab F1 embryos coming from a cross of Kaga males to Cab females. (I) Endogenous *her7-venus* intercept period measurements in hybrid F1 Kaga/Cab reciprocal cross at the 10-11SS shows non-significant difference. Kaga/Cab F1 embryos coming from Kaga females crossed to Cab males have an intercept *her7-venus* period of (53.97 min (SD +/− 2.40)) while Kaga/Cab F1 embryos coming from Kaga males crossed to Cab females have a mean period of (56.00 min (SD +/− 3.27)). Black circle = mean Black line = 95% confidence interval. Welch two sample $t$ test $P = 0.055$. $N = 16$ Kaga/Cab F1 embryos coming from a cross of Kaga females to Cab males, $N = 16$ Kaga/Cab F1 embryos coming from a cross of Kaga males to Cab females.

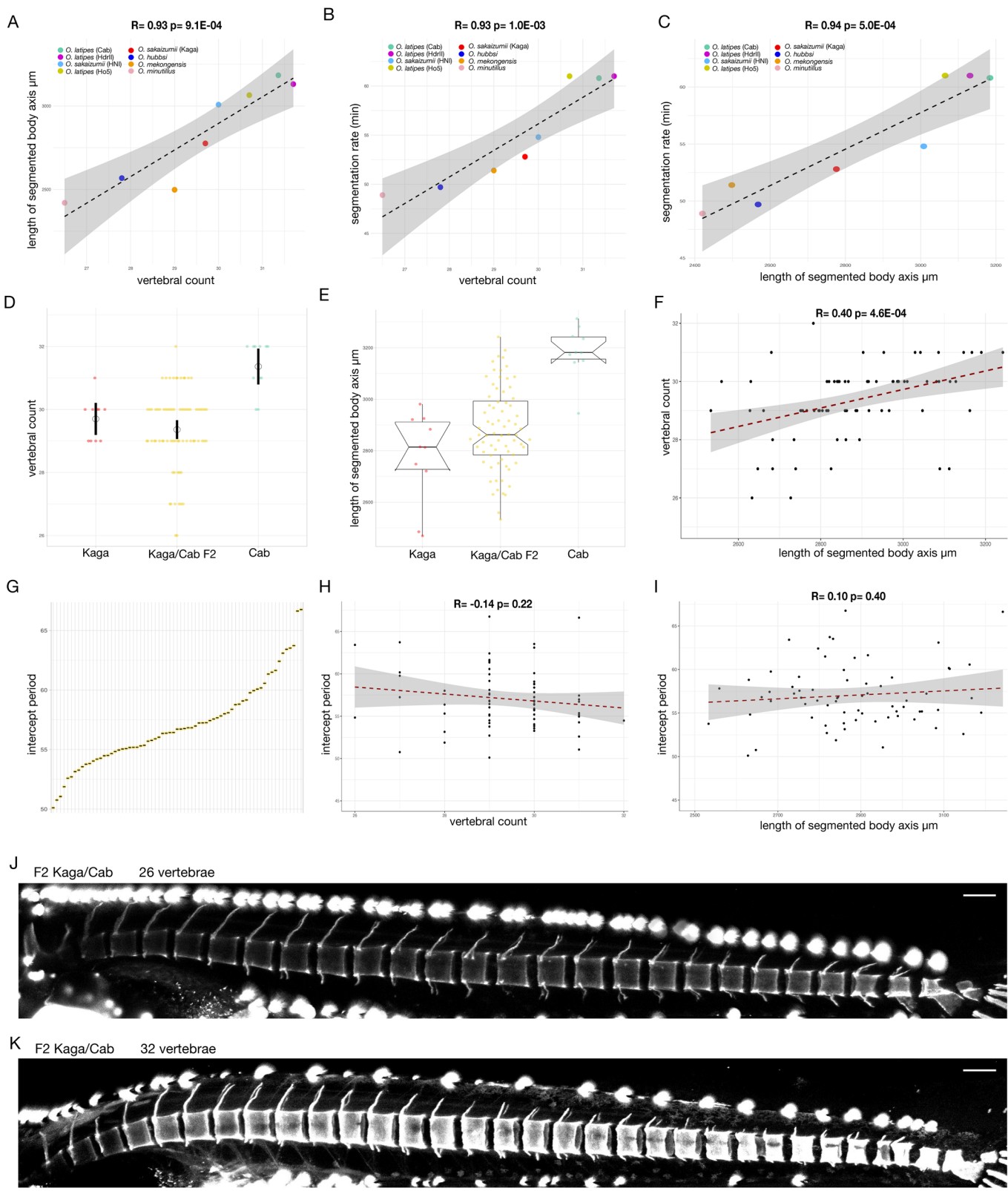

◀ **Figure EV3. Larval size and vertebral count do not correlate with segmentation timing in a subset of F2 embryos.**

(A) Pearson's correlation of average trait values for vertebral count and length of segmented body axis in stage 42 *O. minutillus* (pink), *O. hubbsi* (blue), *O. mekongensis* (orange), HNI (light blue), Kaga (red) and Cab (turquoise), HdrII (magenta), Ho5 (yellow) larvae shows a positive correlation $R = 0.93$ $P$ value = 9.1E-04. Black dotted line= linear fit on average trait values, shaded area = 95% confidence interval. $N = 10$ *O. hubbsi, O. mekongensis*, Kaga, HNI, Ho5, Cab, HdrII larvae $N = 11$ *O. minutillus*. (B) Pearson's correlation of average trait values for vertebral count and segmentation rate in minutes in *O. minutillus* (pink), *O. hubbsi* (blue), *O. mekongensis* (orange), HNI (light blue), Kaga (red) and Cab (aquamarine), HdrII (magenta), Ho5 (yellow) larvae shows a positive correlation $R = 0.93$ $P$ value = 1.0E-03. Black dotted line= linear fit on average trait values, shaded area = 95% confidence interval. $N = 10$ *O. hubbsi, O. mekongensis*, Kaga, HNI, Ho5, Cab, HdrII, $N = 11$ *O. minutillus*. (C) Pearson's correlation of average trait values for length of segmented body axis in stage 42 larvae and segmentation rate in minutes in *O. minutillus* (pink), *O. hubbsi* (blue), *O. mekongensis* (orange), HNI (light blue), Kaga (red) and Cab (aquamarine), HdrII (magenta), Ho5 (yellow) larvae shows a positive correlation $R = 0.94$ $P$ value = 5.0E-04. Black dotted line= linear fit on average trait values, shaded area = 95% confidence interval. $N = 10$ *O. hubbsi, O. mekongensis*, Kaga, HNI, Ho5, Cab, HdrII, $N = 11$ *O. minutillus*. (D) Vertebral count in F2 stage 42 larvae compared to Cab and Kaga F0 populations. Black circle = mean values. Black line = 95% confidence interval. Each dot is one embryo. $N = 72$ F2 Kaga/Cab embryos $N = 10$ F1 Kaga embryos $N = 11$ F0 Cab embryos. (E) Length of segmented body axis in stage 42 F2 larvae compared to Cab and Kaga F0 populations. Black circle = mean values. Black line = 95% confidence interval. Each dot is one embryo. $N = 72$ F2 Kaga/Cab embryos $N = 10$ F1 Kaga embryos $N = 10$ F0 Cab embryos. (F) Pearson's correlation between length of segmented body axis and vertebral count across a subset of F2 Kaga/Cab embryos $R = = 0.4$ $P$ value = 4.6E-04. Red dotted line= linear fit, grey shaded area= 95% confidence interval. $N = 72$ Kaga/Cab F2. (G) Endogenous her7-venus intercept clock period measurements of F2 Kaga/Cab arranged from fastest to slowest. Each black line is an intercept period measurement from one F2 embryo. $N = 70$ F2 Kaga/Cab embryos. (H) Pearson's correlation between vertebral count and intercept clock period across a subset of F2 Kaga/Cab embryos $R = = -0.14$ $P$ value = 0.22. Red dotted line= linear fit, grey shaded area= 95% confidence interval. $N = 70$ Kaga/Cab F2. (I) Pearson's correlation between length of segmented body axis and intercept clock period across a subset of F2 Kaga/Cab embryos $R = = 0.10$ $P$ value = 0.40. Red dotted line= linear fit, grey shaded area= 95% confidence interval. $N = 70$ Kaga/Cab F2. (J, K) ALC bone staining on stage 42 F2 larvae showing the upper and lower bound (26–32) of vertebral counts obtained in the subset of F2 larvae analysed.

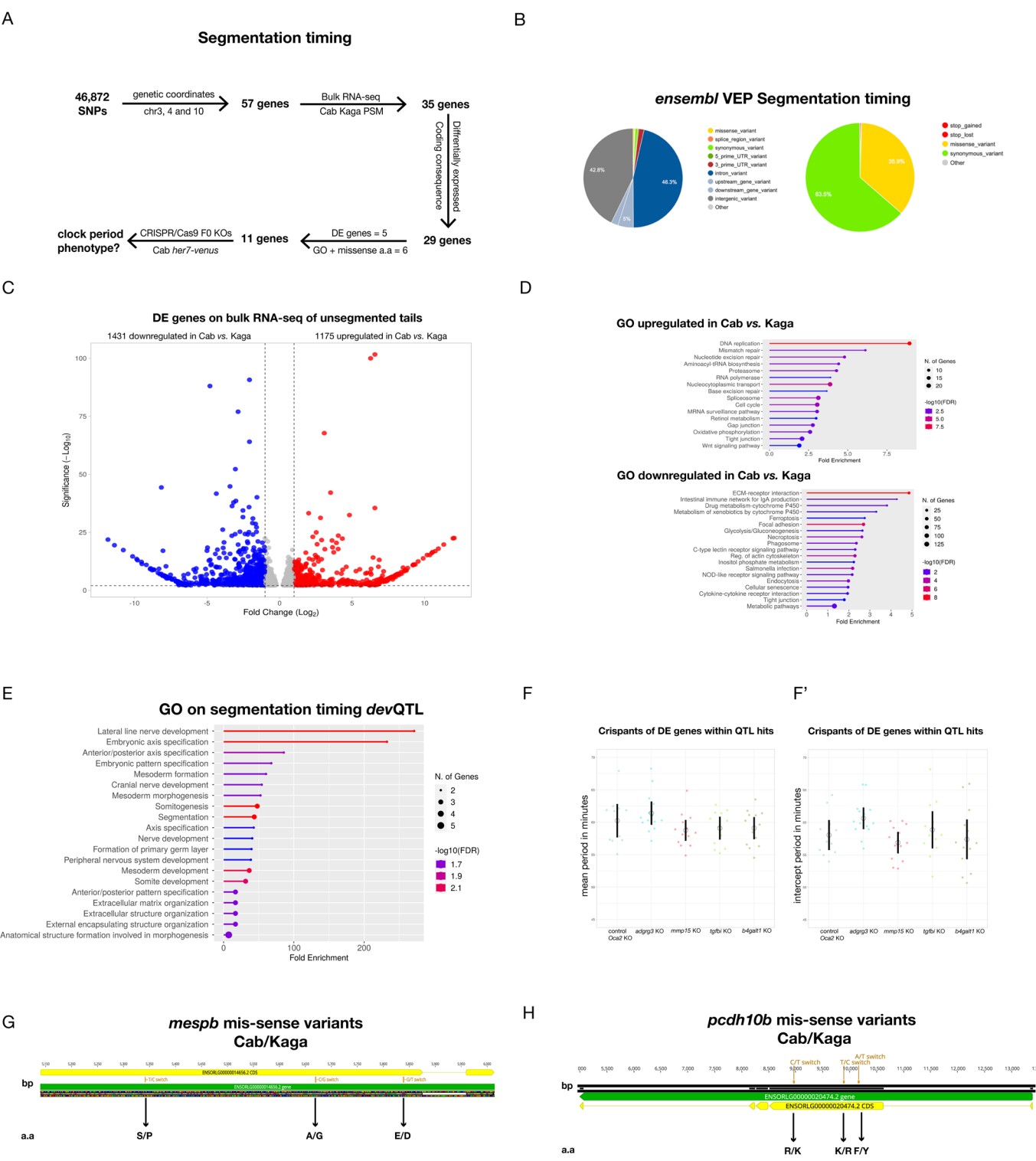

**A** Segmentation timing

**B** *ensembl* VEP Segmentation timing

**C** DE genes on bulk RNA-seq of unsegmented tails

**D** GO upregulated in Cab *vs.* Kaga

GO downregulated in Cab *vs.* Kaga

**E** GO on segmentation timing *dev*QTL

**F** Crispants of DE genes within QTL hits

**F'** Crispants of DE genes within QTL hits

**G** *mespb* mis-sense variants Cab/Kaga

**H** *pcdh10b* mis-sense variants Cab/Kaga

**Figure EV4. *dev*QTL mapping for segmentation timing and bulk-RNA sequencing on Kaga and Cab tails.**

(A) Workflow from *dev*QTL mapping to candidate gene selection. For segmentation timing 46,872 homozygous divergent SNPs between Kaga and Cab were located on chromosomes 3, 4 and 10. Genomic coordinates revealed a total of 57 genes on all 3 chromosomes located in regions that passed the significance threshold. Bulk RNA-sequencing on Kaga and Cab unsegmented PSM narrowed down the number of candidate genes expressed in the PSM to 35 genes. 29 of which were either differentially expressed or contained a coding consequence or both. GO enrichment and candidate gene picking led to a top 11 gene list which were selected to perform F0 CRISPR KOs in *her7-venus* Cab background to check for a clock period phenotype. (B) *ensembl* variant effect predictor (VEP) output for segmentation timing showing the distribution of homozygous divergent SNPs between Kaga and Cab in the F2 Kaga/Cab QTL mapping on segmentation timing, most of the divergent SNPs fall in either intronic or intergenic regions, of the ones that fall within the coding sequence of genes the majority lead to synonymous mutations (63.5%) while only a minority (35.9%) lead to miss-sense mutations. (C) Volcano plot showing differentially expressed (DE) genes on bulk RNA-sequencing of unsegmented Cab and Kaga tails at the 13–14 somite stage. 1431 genes are significantly downregulated in Cab compared to Kaga tails (blue), while 1175 genes are upregulated in Cab compared to Kaga tails (red). (D) Gene Ontology (GO) enrichment categories for upregulated and downregulated genes from the bulk RNA-Sequencing experiment shown in (A). (E) GO analysis on gene list from segmentation timing *dev*QTL mapping that were transcriptionally active in tail tissue (35 genes). (F–F') endogenous *her7-venus* mean and intercept period analysis in F0 Cab Crispants imaged at the 10-11SS on genes differentially expressed and within the *dev*QTL peaks for segmentation timing. Kruskal–Wallis' test $P = 0.2$. (F) Kruskal–Wallis' test $P = 0.05$ (F') $N = 11$ control CRISPR/Cas9 and *Oca2* injected Cab *her7-venus* embryos, $N = 14$ *adgrg3*, $N = 12$ *tgfbi*, $N = 13$ *mmp15*, $N = 29$ *b4galt1*, CRISPR/Cas9 injected into *Cab her7-venus*. (G) Position of mis-sense variants between Cab/Kaga in the coding sequence of *mespb*. Three base-pair (bp) changes cause 3 amino acid (a.a) changes: Serine/Proline (S/P), Alanine/Glycine (A/G) and Glutamic Acid/Aspartic Acid (E/D) are highlighted. Visualisation done using *Geneious*. (H) Position of mis-sense variants between Cab/Kaga in the coding sequence of *pcdh10b*. Three base-pair (bp) changes cause 3 amino acid (a.a) changes: Arginine/Lysine (R/K), Lysine/Arginine (K/R) and Phenylalanine/Tyrosine (F/Y) are highlighted. Visualisation using *Geneious*.

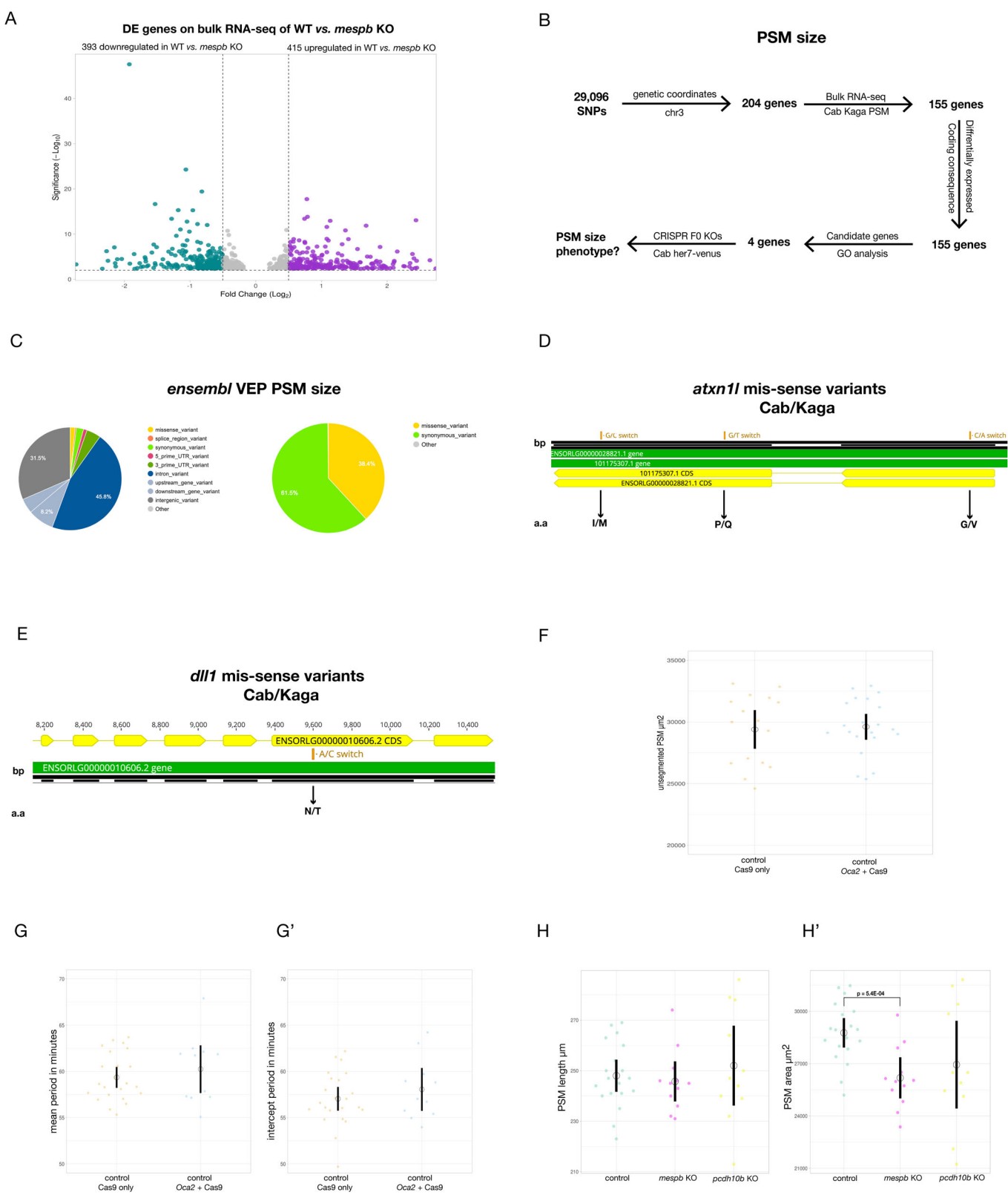

◀  **Figure EV5.  Bulk RNA-sequencing on Cab *wild-type* vs. *mespb* KO tails and *dev*QTL mapping on PSM size.**

(**A**) Volcano plot showing differentially expressed genes on bulk RNA-sequencing of Cab *wild-type* and *mespb* Crispant tails at the 13–14 somite stage. 393 genes are significantly downregulated in *wild-type* compared to *mespb* Crispant tails (green), while 415 genes are upregulated in *wild-type* compared to *mespb* Crispant (magenta). (**B**) Workflow from *dev*QTL mapping to candidate gene selection. For PSM size 29,096 homozygous divergent SNPs between Kaga and Cab are located on chromosomes 3. Genomic coordinates revealed a total of 204 genes located in regions that passed the significance threshold. Bulk RNA-sequencing on Kaga and Cab unsegmented PSM showed 155 genes transcriptionally acitve. GO annotation and candidate gene picking led to a top 4 genes which were selected to perform F0 CRISPR/Cas9 KOs in Cab background to assess PSM size. (**C**) VEP output for PSM size showing the distribution of homozygous divergent SNPs between Kaga and Cab in the F2 Kaga/Cab QTL on PSM size, majority of the divergent SNPs fall in either intronic or intergenic regions, of the ones that fall within the coding sequence of genes the majority lead to synonymous mutations (61.5%) while only a minority (38.4%) lead to miss-sense mutations. (**D**) position of mis-sense variants between Cab/Kaga in the coding sequence of *atxn1l*. Three base-pair (bp) changes cause 3 amino acid (a.a) changes: Isoleucine/Methionine (I/M), Proline/Glutamine (P/Q) and Glycine/Valine (G/V) are highlighted. Visualization using *Geneious*. (**E**) position of mis-sense variants between Cab/Kaga in the coding sequence of *dll1*. One base-pair (bp) change causes 1 amino acid (a.a) change: Asparagine/Threonine (N/T) is highlighted. Visualisation using *Geneious*. (**F**) Comparison of PSM size for *Oca2* + Cas9 injected control embryos as opposed to Cas9 only injected control embryos (shown in Fig. 4). Welch two sample *t* test $P = 0.80$. $N = 11$ *Oca2* + Ca9, $N = 23$ Cas9 only (**G–G′**) Comparison of endogenous *her7-venus* mean and intercept period values for *Oca2* + Cas9 injected control embryos (shown in EV4F-F′) as opposed to Cas9 only injected control embryos (shown in Fig. 4; Fig. S4A,B) Welch two sample *t* test $P = 0.48$ (**G**) and $P = 0.39$ (**G′**) $N = 11$ *Oca2* + Ca9, $N = 23$ Cas9 only. (**H–H′**) PSM size in *mespb* and *pcdh10b* Crispants compared to control embryos. Welch two sample *t* test PSM length *mespb* KO $P = 0.60$, *pcdh10b* KO $P = 0.59$. Welch two sample *t* test PSM area *mespb* KO $P = 5E{-}04$, *pcdh10b* KO $P = 0.13$. $N = 19$ control embryos, $N = 13$ *mespb* KO Crispants, $N = 11$ *pcdh10b* KO Crispants.

