## [Peer Review File · The EMBO Journal]

Modular control of vertebrate axis segmentation in time and space

Ali Seleit, Ian Brettell, Tomas Fitzgerald, Carina Vibe, Felix Loosli, Joachim Wittbrodt, Kiyoshi Naruse, Ewan Birney, and Alexander Aulehla

Corresponding authors: Alexander Aulehla (aulehla@embl.de) , Ewan Birney (birney@ebi.ac.uk)

Review Timeline:

Transferred from Review Commons:	22nd Apr 24
Editorial Decision:	31st May 24
Revision Received:	24th Jun 24
Accepted:	11th Jul 24

Editor: Ieva Gailite

Transaction Report:

This manuscript was transferred to The EMBO Journal following peer review at Review Commons.

Review #1

1. Evidence, reproducibility and clarity:

Evidence, reproducibility and clarity (Required)

Seleit and colleagues set out to explore the genetics of developmental timing and tissue size by mapping natural genetic variation associated with segmentation clock period and presomitic mesoderm (PSM) size in different species of Medaka fish. They first establish the extent of variation between five different Medaka species of in terms of organismal size, segmentation rate, segment size and presomitic mesoderm size, among other traits. They find that these traits are species-specific but strongly correlated. In a massive undertaking, they then perform developmental QTL mapping for segmentation clock period and PSM size in a set of ~600 F2 fish resulting from the cross of *Orizyas sakaizumii* (Kaga) and *Orizyas latipes* (Cab). Correlation between segmentation period and segment size was lost among the F2s, indicating that distinct genetic modules control these traits. Although the researchers fail to identify causal variants driving these traits, they perform proof of concept perturbations by analyzing F0 Crispants in which candidate genes were knocked out. Overall, the study introduces a completely new methodology (QTL mapping) to the field of segmentation and developmental tempo, and therefore provides multiple valuable insights into the forces driving evolution of these traits.

****Major comments:****

- The first sentence in the abstract reads "How the timing of development is linked to organismal size is a longstanding question". It is therefore disappointing that organismal size is not reported for the F2 hybrids. Was larval length measured in the F2s? If so, it should be reported. It is critical to understand whether the correlation between larval size and segmentation clock period is preserved in F2s or not, therefore determining if they represent a single or separate developmental modules. If larval length data were not collected, the authors need to be more careful with their wording. In the current version of the paper, organismal size is often incorrectly equated to tissue size (e.g. PSM size, segment size). For example, in page 3 lines 33-34, the authors state that faster segmentation occurred in embryos of smaller size (Fig. 1D). However, Fig. 1D shows correlation between segmentation rate and unsegmented PSM area. The appropriate data to show would be segmentation rate vs. larval or adult length.
- Is my understanding correct in that the *her7-venus* reporter is carried by the Cab F0 but not the Kaga F0? Presumably only F2s which carried the reporter were selected for phenotyping. I would expect the location of the reporter in the genome to be obvious in Figure 3J as a region that is only Cab or het but never Kaga. Can the authors please point to the location of the reporter?
- devQTL mapping in this study seems like a wasted opportunity. The authors perform mapping only to then hand pick their targets based on GO annotations. This biases the study towards genes known to be involved in PSM development, when part of the appeal of QTL mapping is precisely its unbiased nature and the potential to discover new functionally relevant genes. The authors need to better justify their rationale for candidate prioritization from devQTL peaks. The GO analysis should be shown as supplemental data. What criteria were used to select genes based on GO annotations?

- Analysis of the predicted functional consequence of divergent SNPs (Fig. S6B, F) is superficial. Among missense variants, which genes harbor the most deleterious mutations? Which missense variants are located in highly conserved residues? Which genes carry variants in splice donors/acceptors? Carefully assessing the predicted effect of SNPs in coding regions would provide an alternative, less biased approach to prioritize candidate genes.
- Another potential way to prioritize candidate genes within devQTL peaks would be to use the RNA seq data. The authors should perform differential expression analysis between Kaga and Cab RNA-seq datasets. Do any of the differentially expressed genes fall within the devQTL peaks?
- The use of crisprants to functionally test candidate genes is inappropriate. Crisprants do not mimic the effect of divergent SNPs and therefore completely fail to prove causality. While it is completely understandable that Medaka fish are not amenable to the creation of multiple knock-in lines where divergent SNPs are interconverted between species, better justification is needed. For instance, is there enough data to suggest that the divergent alleles for the candidate genes tested are loss of function? Why was a knockout approach chosen as opposed to overexpression?
- Along the same line, now that two candidate genes have been shown to modulate the clock period in crisprants (*mespb* and *pcdh10b*), the authors should at least attempt to knock in the respective divergent SNPs for one of the genes. This is of course optional because it would imply several months of work, but it would significantly increase the impact of the study.

****Minor Comments****

- It would be highly beneficial to describe the ecological differences between the two Medaka species. For example, do the northern *O. sakaizumii* inhabit a colder climate than the southern *O. latipes*? Is food more abundant or easily accessible for one species compared to the other? What, if anything, has been described about each species' ecology?
- The authors describe two different methods for quantifying segmentation clock period (mean vs. intercept). It is still unclear what is the difference between Figs. 3A (clock period), S4A (mean period) and S4B (intercept period). Is clock period just mean period? Are the data then shown twice? How do Fig. 3A and S4A differ?
- devQTL as shorthand for developmental QTL should be defined in page 4 line 1 (where the term first appears), not later in line 12 of the same page.
- Python code for period quantification should be uploaded to Github and shared with reviewers.
- RNA-seq data should be uploaded to a publicly accessible repository and the reviewer token shared with reviewers.
- Why are the maintenance (27-28C) vs. imaging (30C) temperatures different?
- For Crisprants, control injections should have included a non-targeting sgRNA control instead of simply omitting the sgRNA.
- It is difficult to keep track of the species and strains. It would be most helpful if Fig. S1 appeared instead in main figure 1.

2. Significance:

Significance (Required)

- The study introduces a new way of thinking about segmentation timing and size scaling by considering natural variation in the context of selection. This new framing will have an important impact on the field.
- Perhaps the most significant finding is that the correlation between segment timing and size in wild populations is driven not by developmental constraints but rather selection pressure, whereas segment size scaling does form a single developmental module. This finding should be of interest to a broad audience and will influence how researchers in the field approach future studies.
- It would be helpful to add to the conclusion the author's opinion on whether segmentation timing is a quantitative trait based on the number of QTL peaks identified.
- The authors should be careful not to assign any causality to the candidate genes that they test in crispants.
- The data and results are generally well-presented, and the research is highly rigorous.
- Please note I do have the expertise to evaluate the statistical/bioinformatic methods used for devQTL mapping.

3. How much time do you estimate the authors will need to complete the suggested revisions:

Estimated time to Complete Revisions (Required)

(Decision Recommendation)

Between 1 and 3 months

No

Review #2

1. Evidence, reproducibility and clarity:

Evidence, reproducibility and clarity (Required)

Seleit et al. investigate the correlation between segment size, presomitic mesoderm and the rhythm of periodic oscillations in the segmentation clock of developing medaka fish. Specifically, they aim to identify the genetic determinants for said traits. To do so, they employ a common garden approach and measure such traits in separate strains (F0) and in interbreedings across two generations (F1 and F2). They find that whereas presomitic mesoderm and segment size are genetically coupled, the tempo of her7 oscillations it is not. Genetic mapping of the F0 and F2 progeny allows them to identify regions associated to said traits. They go on and perturb 7 loci associated to the segmentation clock and X related to segment size. They show that 2/7 have a tempo defect, and 2/ affect size.

****Major comments:****

The conclusions are convincing and well supported by the data. I think the work could be published as is in its current state, and no additional experiments that I can think of are needed to support the claims in the paper.

****Minor comments:****

- The authors could provide a more detailed characterization of the identified SNPs associated to the clock and to PSM size. For the segmentation clock, the authors identify 46872 SNPs, most of which correspond to non-coding regions and are associated to 57 genes. They narrow down their approach to those expressed in the PSM of Cab Kaga. Was the RNA selected from F1 hybrids? I wonder if this would impact the analysis for tempo and or size in any way, as F2 are derived from these, and they show broader variability in the clock period than the F0 and F1 fishes.
- It would be good if the authors could discuss if there were any associated categories or overall functional relationships between the SNPs/genes associated to size. And what about in the case of timing?
- Have any of the candidate genes or regulatory loci been associated to clock defects (57) or segment size (204) previously in the literature?
- When the authors narrow down the candidate list, it is not clear if the genes selected as expressed in the PSM are tissue specific. If they are, I wonder if genes with ubiquitous expression would be more informative to investigate tempo of development more broadly. It would be good if the authors could specifically discuss this point in the manuscript.
- Can the authors speculate mechanistically why mespb or pchd10b accelerates the period of her7 oscillations?
- Are there any size difference associated to the functionally validated clock mutants?
- Ref 27 shows a lack of correlation between body size and the segmentation period in various species of mammals. The work supports their findings, and it would be good to see this discussed in the text.

2. Significance:

Significance (Required)

The work is quite remarkable in terms of the multigenerational genetic analysis performed. The authors have analysed >600 embryos from three separate generations to obtain quantitative data to answer their question (herculean task!). Moreover, they have associated this characterization to specific SNPs. Then, to go beyond the association, they have generated mutant lines and identified specific genes associated to the traits they set out to decipher.

To my knowledge, this is the first project that aims to identify the genetic determinants for developmental timing. Recent work on developmental timing in mammals has focused on interspecies comparisons and does not provide genetic evidence or insight into how tempo is regulated in the genome. As for vertebrates, recent work from zebrafish has profiled temperature effects on cell proportions and developmental timing. However, the genetic approach of this work is quite elegant and neat.

Conceptually, it is quite important and unexpected that overall size and tempo are not related. Body size, lifespan, basal metabolic rates and gestational period correlate positively and we tend to think that mechanistically they would all be connected to one another. This paper and Lazaro et al. 2023 (ref 27) are one of the first in which this preconception is challenged in a very methodical and conclusive manner. I believe the work is a breakthrough for the field and this work would be interesting for the field of biological timing, for the segmentation clock community and more broadly for all developmental biologists.

My field is quantitative stem cell biology and I work on developmental timing myself, so I acknowledge that I am biased in the enthusiasm for the work. It should be noted that as an expert on the field, I have identified instances where other work hasn't been as insightful or well developed in comparison to this piece. It is also worth noting that I am not an expert in fish development, phylogenetic studies or GWAS analyses, so I am not capable to assess any pitfalls in that respect.

3. How much time do you estimate the authors will need to complete the suggested revisions:

Estimated time to Complete Revisions (Required)

(Decision Recommendation)

Less than 1 month

Yes

Review #3

1. Evidence, reproducibility and clarity:

Evidence, reproducibility and clarity (Required)

****Summary:****

This manuscript explores the temporal and spatial regulation of vertebrate body axis development and patterning. In the early stages of vertebrate embryo development, the axial mesoderm (presomitic mesoderm - PSM) undergoes segmentation, forming structures known as somites. The exact genetic regulation governing somite and PSM size, and their relationship to the periodicity of somite formation remains unclear.

To address this, the authors used two evolutionarily closely related Medaka species, *Oryzias sakaizumii* and *Oryzias latipes*, which, although having distinct characteristics, can produce viable offspring. Through analysis spanning parental (generation F0) and offspring (generations F1 and F2) generations, the authors observed a correlation between PSM and somite size. However, they found that size scaling does not correlate with the timing of somitogenesis.

Furthermore, employing developmental quantitative trait loci (devQTL) mapping, the authors identified several new candidate loci that may play a role during somitogenesis, influencing timing of segment formation or segment size. The significance of these loci was confirmed through an innovative CRISPR-Cas9 gene editing approach.

This study highlights that the spatial and temporal aspects of vertebrate segmentation are independently controlled by distinct genetic modular mechanisms.

****Major comments:****

1. In the main text page 3, lines 11 and 12, the authors state that the periodicity of the embryo clock of the F1 generation is the intermediate between the parental F0 lineages. However, the authors look only at the periodicity of the Cab strain (*Oryzias latipes*) segmentation clock. The authors should have a reporter fish line for the Kaga strain (*Oryzias sakaizumii*) to compare the segmentation clock of both parental strains and their offspring. Since it could be time consuming and laborious, I advise to alternatively rephrase the text of the manuscript.
2. It is evident that only a few F0 and F1 animals were analyzed in comparison with the F2 generation. Could the authors kindly explain whether and how this could bias or skew the observed results?
3. It would be interesting to create fish lines with the validated CRISPR-Cas9 gene manipulations in different genetic contexts (Cab or Kaga) to analyze the true impact on the segmentation clock and/or PSM & somite sizes.

4. Please add the results of the Go Analysis as supplementary material.

****Minor comments:****

1. In the main text, page 2, line 29, Supplementary Figure 1D should be referenced.
2. In the main text, page 2, line 32, the authors refer to Figure 1B, but it should be 1C.
3. Regarding the topic "Correlation of segmentation timing and size in the *Oryzias* genus" the authors should also give information on the total time of development of the different *Oryzias* species, as well as the total number of formed somites.
4. In Figures 3A and B, please add info on the F1 lines for comparison.
5. Supplementary Figures 2F shows that the generation F1 PSM is similar to Cab F0, and not an intermediate between Kaga F0 and Cab F0. This is interesting and should be discussed.
6. Supplementary Figures 6C to H are not mentioned either in the main text or in the extended information. Please add/mention accordingly.
7. The order of Supplementary Figure 8 E to H and A to D appears to be not correct and not following the flow of the text. Please update/correct accordingly.
8. The authors should choose between "Fig.", "Fig", "fig.", "fig" or "Figure". All 'variants' can be found in the text.
9. The color scheme of several figures (graphs with colored dots) should be revised. Several appear to be difficult to discern and analyze.
10. Please address/discuss following questions: What are the known somitogenesis regulating genes in Medaka? How do they correlate with the new candidates?

2. Significance:

Significance (Required)

General assessment:

This interesting manuscript describes a novel approach to study and find new players relevant to the regulation of vertebrate segmentation. By employing this innovative methodology, the authors could elegantly demonstrate that the segmentation clock periodicity is independent from the sizes of the PSM and forming somites. The authors were further able to find new genes that may be involved in the regulation of the segmentation clock periodicity and/or the size of the PSM & somites. A limitation of this study is the fact that the results mainly rely on differences between the two species. The integration of additional Medaka species would be beneficial and may help uncover relevant genes and genetic contexts.

Advance:

To my best knowledge this is the first time that such a methodology was employed to study the segmentation clock and axial development. Although the topic has been extensively studied in several model organisms, such as mice, chicken, and zebrafish, none of them correlated the size of the embryonic tissues and the periodicity of the embryo clock. This study brings novel technological and functional advances to the study of vertebrate axial development.

Audience:

This work is particularly interesting to basic researchers, especially in the field of developmental biology and represents a fresh new approach to study a core developmental process. This study further opens the exciting possibility of using a similar methodology to investigate other aspects of vertebrate development. It is a timely and important manuscript which could be of interest to a wider scientific audience and readership.

3. How much time do you estimate the authors will need to complete the suggested revisions:

Estimated time to Complete Revisions (Required)

(Decision Recommendation)

Between 3 and 6 months

Yes

Full Revision

Manuscript number: RC-2023-02279R

Corresponding author(s): Alexander, Aulehla

1. General Statements

We provide here a full revision addressing all the points raised by the reviewers and also include in the revised manuscript a number of additional experiments.

Reviewer #1

Evidence, reproducibility and clarity

Seleit and colleagues set out to explore the genetics of developmental timing and tissue size by mapping natural genetic variation associated with segmentation clock period and presomitic mesoderm (PSM) size in different species of Medaka fish. They first establish the extent of variation between five different Medaka species in terms of organismal size, segmentation rate, segment size and presomitic mesoderm size, among other traits. They find that these traits are species-specific but strongly correlated. In a massive undertaking, they then perform developmental QTL mapping for segmentation clock period and PSM size in a set of ~600 F2 fish resulting from the cross of *Orizyas sakaizumii* (Kaga) and *Orizyas latipes* (Cab). Correlation between segmentation period and segment size was lost among the F2s, indicating that distinct genetic modules control these traits. Although the researchers fail to identify causal variants driving these traits, they perform proof of concept perturbations by analyzing F0 Crispants in which candidate genes were knocked out. Overall, the study introduces a completely new methodology (QTL mapping) to the field of segmentation and developmental tempo, and therefore provides multiple valuable insights into the forces driving evolution of these traits.

Major comments:

- The first sentence in the abstract reads "How the timing of development is linked to organismal size is a longstanding question". It is therefore disappointing that organismal size is not reported for the F2 hybrids. Was larval length measured in the F2s? If so, it should be reported. It is critical to understand whether the correlation between larval size and segmentation clock period is preserved in F2s or not, therefore determining if they represent a single or separate developmental modules. If larval length data were not collected, the authors need to be more careful with their wording.

The question the reviewer raises here is indeed a very relevant one, and a question that we also were curious about ourselves. While it was not possible (logistically) to grow the 600 F2 fish to adulthood, we did measure larval length in a subset of F2 hatchling (n=72) to ask precisely the question the reviewer raises here. Our results (new Supplementary Figure 5) show that the correlation between larval length and segmentation timing (which we report across the *Orizyas* species) is absent in the F2s. This indeed argues that the traits represent separate developmental modules.

Full Revision

In the current version of the paper, organismal size is often incorrectly equated to tissue size (e.g. PSM size, segment size). For example, in page 3 lines 33-34, the authors state that faster segmentation occurred in embryos of smaller size (Fig. 1D). However, Fig. 1D shows correlation between segmentation rate and unsegmented PSM area. The appropriate data to show would be segmentation rate vs. larval or adult length.

The reviewer is correct. We have now linked the data more clearly to data we show in Supplementary Figure 1, which shows that adult length and adult mass are strongly correlated (S1A) and that adult mass is in turn strongly correlated with segmentation rate in the different *Oryzias* species (S1B). Additionally main Figure 1B shows that larval length is correlated with PSM length. We have corrected the main text to reflect these relationships more clearly.

- Is my understanding correct in that the *her7-venus* reporter is carried by the Cab F0 but not the Kaga F0? Presumably only F2s which carried the reporter were selected for phenotyping. I would expect the location of the reporter in the genome to be obvious in Figure 3J as a region that is only Cab or het but never Kaga. Can the authors please point to the location of the reporter?

The reviewer is correct. Indeed the location of our *her7-venus* KI is on chromosome 16 and the recombination patterns on this chromosome overwhelmingly show either Hom Cab (green) or Het Cab/Kaga (Black). This is expected as we selected fish carrying the *her7-venus* KI for phenotyping.

- devQTL mapping in this study seems like a wasted opportunity. The authors perform mapping only to then hand pick their targets based on GO annotations. This biases the study towards genes known to be involved in PSM development, when part of the appeal of QTL mapping is precisely its unbiased nature and the potential to discover new functionally relevant genes. The authors need to better justify their rationale for candidate prioritization from devQTL peaks. The GO analysis should be shown as supplemental data. What criteria were used to select genes based on GO annotations?

We have now commented on these valid points and outlined our rationale in more detail in the text (page 4, lines 20-30). Our rationale now also includes selection of differentially expressed genes (n=5 genes) that fall within segmentation timing devQTL hits (for more details see below). Essentially, while we indeed finally focused on the proof of principle using known genes, these genes were previously not known to play a role in either setting the timing of segmentation or controlling the size of the PSM. Hence, we do think our strategy demonstrates the “the potential to discover new functionally relevant genes”, even though the genes themselves had been involved overall in somitogenesis. We added the GO analysis as supplemental data as requested (new Supplementary Figure 7E).

- Analysis of the predicted functional consequence of divergent SNPs (Fig. S6B, F) is superficial. Among missense variants, which genes harbor the most deleterious mutations? Which missense variants are located in highly conserved residues? Which genes carry variants in splice donors/acceptors? Carefully assessing the predicted effect of SNPs in coding regions would provide an alternative, less biased approach to prioritize

Full Revision

candidate genes.

We now included our analysis of SNPs based on the Variant effect predictor (VEP) tool from *ensembl*. This analysis does rank the predicted severity of the SNP on protein structure and function (Impact: low, moderate, high) and does annotate which variants can affect splice donors/acceptors. The VEP analysis for both phenotypes is now added to the manuscript as supplemental data (new Supplementary Data S2, S5).

- Another potential way to prioritize candidate genes within devQTL peaks would be to use the RNA seq data. The authors should perform differential expression analysis between Kaga and Cab RNA-seq datasets. Do any of the differentially expressed genes fall within the devQTL peaks?

As suggested we have performed this additional experiment and report the RNAseq differential analysis in new Supplement Figure 7C-D. The analysis revealed 2606 differentially expressed genes in the PSM between Kaga and Cab, five of which were candidate genes from the devQTL analysis. We now tested all of these (5 in total, 4 new and 1 previously targeted *adgrg1*) for segmentation timing by CRISPR/Cas9 KO in the *her7-venus* background, none of which showed a timing phenotype (new Supplementary Figure 7F-F'). We provide the complete set of results in new Supplementary Figure 7, Supplementary Data file 3 (DE-genes), all data were deposited on publicly available repository Biostudies under accession number: E-MTAB-13927.

- The use of crisprants to functionally test candidate genes is inappropriate. Crisprants do not mimic the effect of divergent SNPs and therefore completely fail to prove causality. While it is completely understandable that Medaka fish are not amenable to the creation of multiple knock-in lines where divergent SNPs are interconverted between species, better justification is needed. For instance, is there enough data to suggest that the divergent alleles for the candidate genes tested are loss of function? Why was a knockout approach chosen as opposed to overexpression?

We agree with the reviewer that we do not address the causality of SNPs with the CRISPR/Cas9 KO approach we followed. And medaka does offer the genome editing capabilities to create tailored sequence modifications. So in principle, this can be done. In practice, however, we reasoned that any given SNP will contribute only partially to the observed phenotypes and combinatorial sequence edits are simply very laborious given the current state of the art in genome editing technologies. We therefore opted for an alternative proof of principle approach that aims to “to discover new functionally relevant genes”, not SNPs.

- Along the same line, now that two candidate genes have been shown to modulate the clock period in crisprants (*mespb* and *pcdh10b*), the authors should at least attempt to knock in the respective divergent SNPs for one of the genes. This is of course optional because it would imply several months of work, but it would significantly increase the impact of the study.

As above, this is in principle the correct rationale to follow though very time, cost and labour intensive. It is for the later practical consideration that we decided not to follow this option.

Minor Comments

- It would be highly beneficial to describe the ecological differences between the two Medaka species. For example, do the northern *O. sakaizumii* inhabit a colder climate than the southern *O. latipes*? Is food more

Full Revision

abundant or easily accessible for one species compared to the other? What, if anything, has been described about each species' ecology?

There are indeed differences in the ecology of both species, with the northern *O. sakaizumii* inhabiting a colder climate than the southern *O. latipes*. In addition, it is known that the breeding season is shorter in the north than the south, and also there is the fact that northern species have been shown to have a faster juvenile growth rate than southern species. While it would be premature to link those ecological factors to the timing differences we observe, we can certainly speculate. A line to this effect has been added to the main text (Page 5, line 28-30).

- The authors describe two different methods for quantifying segmentation clock period (mean vs. intercept). It is still unclear what is the difference between Figs. 3A (clock period), S4A (mean period) and S4B (intercept period). Is clock period just mean period? Are the data then shown twice? How do Fig. 3A and S4A differ?

The clock period shown in all the main figures is the intercept period, which was also used for the devQTL analysis. Both measurements (mean and intercept) are indeed highly correlated and we include both in supplement for completeness.

- devQTL as shorthand for developmental QTL should be defined in page 4 line 1 (where the term first appears), not later in line 12 of the same page.

Noted and corrected, we thank the reviewer for spotting this error.

- Python code for period quantification should be uploaded to Github and shared with reviewers.

All period quantification code that was used in this study was obtained from the publicly available tool Pyboat (<https://www.biorxiv.org/content/10.1101/2020.04.29.067744v3>). All code that is used in PyBoat is available from the Github page of the creator of the tool (<https://github.com/tensionhead/pyBOAT>). Both are linked in the references and materials and methods sections.

- RNA-seq data should be uploaded to a publicly accessible repository and the reviewer token shared with reviewers.

We have uploaded all RNA-sequencing Data to public repository BioStudies under accession numbers : E-MTAB-13927, E-MTAB-13928. This information is now also added to material and methods in the manuscript text.

- Why are the maintenance (27-28C) vs. imaging (30C) temperatures different?

Full Revision

Medaka fish have a wide range of temperatures they can physiologically tolerate, i.e. 17-33. The temperature 30C was chosen for practical reasons, i.e. a slightly faster developmental rate enables higher sample throughput in overnight real-time imaging experiments.

- For Crispants, control injections should have included a non-targeting sgRNA control instead of simply omitting the sgRNA.

We agree a non-targeting sgRNA control can be included, though we choose a different approach. For clarity, we now also include a control targeting *Oca2*, a gene involved in the pigmentation of the eye to probe for any injection related effect on timing and PSM size. As expected, 3 sgRNAs + Cas9 against *Oca2* had no impact on timing or PSM size. This data is now shown in new Supplementary Figure 9 F-G'.

- It is difficult to keep track of the species and strains. It would be most helpful if Fig. S1 appeared instead in main figure 1.

We agree and included an overview of the phylogenetic relationship of all species and their geographical locales in new Figure 1 A-B.

Significance

- The study introduces a new way of thinking about segmentation timing and size scaling by considering natural variation in the context of selection. This new framing will have an important impact on the field.

- Perhaps the most significant finding is that the correlation between segment timing and size in wild populations is driven not by developmental constraints but rather selection pressure, whereas segment size scaling does form a single developmental module. This finding should be of interest to a broad audience and will influence how researchers in the field approach future studies.

- It would be helpful to add to the conclusion the author's opinion on whether segmentation timing is a quantitative trait based on the number of QTL peaks identified.

- The authors should be careful not to assign any causality to the candidate genes that they test in crispants.

- The data and results are generally well-presented, and the research is highly rigorous.

- Please note I do have the expertise to evaluate the statistical/bioinformatic methods used for devQTL mapping.

Full Revision

Reviewer #2

Evidence, reproducibility and clarity

Seleit et al. investigate the correlation between segment size, presomitic mesoderm and the rhythm of periodic oscillations in the segmentation clock of developing medaka fish. Specifically, they aim to identify the genetic determinants for said traits. To do so, they employ a common garden approach and measure such traits in separate strains (F0) and in interbreedings across two generations (F1 and F2). They find that whereas presomitic mesoderm and segment size are genetically coupled, the tempo of *her7* oscillations it is not. Genetic mapping of the F0 and F2 progeny allows them to identify regions associated to said traits. They go on an perturb 7 loci associated to the segmentation clock and X related to segment size. They show that 2/7 have a tempo defect, and 2/ affect size.

Major comments:

The conclusions are convincing and well supported by the data. I think the work could be published as is in its current state, and no additional experiments that I can think of are needed to support the claims in the paper.

Minor comments:

- The authors could provide a more detailed characterization of the identified SNPs associated to the clock and to PSM size. For the segmentation clock, the authors identify 46872 SNPs, most of which correspond to non-coding regions and are associated to 57 genes. They narrow down their approach to those expressed in the PSM of Cab Kaga. Was the RNA selected from F1 hybrids? I wonder if this would impact the analysis for tempo and or size in any way, as F2 are derived from these, and they show broader variability in the clock period than the F0 and F1 fishes.

The RNA was obtained from the pure F0 strains and we have now extended this analysis by deep bulk-RNA sequencing and differential gene expression analysis. As indicated also to reviewer 1, this revealed 2606 differentially expressed genes in the unsegmented tails of Kaga and Cab embryos, some of which occurred in devQTL peaks. Based on this information we expanded our list of CRISPR/Cas9 KOs by targeting all differentially expressed genes (5 in total, 4 new and 1 previously targeted) for segmentation timing, none of which showed a timing phenotype (new Supplementary figure 7C-D). We provide the complete set of results in new Supplementary Figure 7, Supplementary Data file 3 (DE-genes). All data were deposited on publicly available repository Biostudies under accession number: E-MTAB-13927.

- It would be good if the authors could discuss if there were any associated categories or overall functional relationships between the SNPs/genes associated to size. And what about in the case of timing?

In the case of PSM size there were no clear GO terms or functional relationships between the genes that passed the significance threshold on chromosome 3.

For the 35 genes related to segmentation timing, there were a number of GO enrichment terms directly related to somitogenesis. We have included the GO analysis in the new Supplementary Figure 7E.

- Have any of the candidate genes or regulatory loci been associated to clock defects (57) or segment size (204) previously in the literature?

Full Revision

To the best of our knowledge none of the genes have been associated with clock or PSM size defects so far. It might be worthwhile using our results to probe their function in other systems enabling higher throughput functional analysis, such as newly developed organoid models.

- When the authors narrow down the candidate list, it is not clear if the genes selected as expressed in the PSM are tissue specific. If they are, I wonder if genes with ubiquitous expression would be more informative to investigate tempo of development more broadly. It would be good if the authors could specifically discuss this point in the manuscript.

We have not addressed the spatial expression pattern of the 35 identified PSM genes in this study, so we cannot speculate further. But the reviewer raises an important point, how timing of individual processes (body axis segmentation) are linked at organismal scale is indeed a fundamental, additional, question that will be addressed in future studies, indeed the *in-vivo* context we follow here would be ideal for such investigations.

- Can the authors speculate mechanistically why *mespb* or *pchd10b* accelerates the period of *her7* oscillations?

While we do not have a mechanistic explanation yet, an additional experiment we performed, i.e. bulk-RNAsequencing on WT and *mespb* mutant tails, provided additional insight, we now added this data to the manuscript. This analysis revealed 808 differentially expressed genes between *wt* and *mespb* mutants. Interestingly, many of these affected genes are known to be expressed outside of the *mespb* domain, i.e. in the most posterior PSM (i.e. *tbxt*, *foxb1*, *tbx6*, *axin2*, *fgf8*, amongst others). This indicates that the effect of *mespb* downregulation is widespread and possibly occurs at an earlier developmental stage. This requires more follow up studies. This data is now shown in new Supplementary figure 9A, Supplementary Data file S4. We now comment on this point in the revised manuscript.

- Are there any size difference associated to the functionally validated clock mutants?

We addressed this point directly and added this analysis as supplementary Figure 9H-H'. While *pcdh10b* mutants do not show any detectable difference in PSM size, we find a small, statistically significant reduction in PSM size (area but not length) in *mespb* mutants. All this data is now included in the revised manuscript.

- Ref 27 shows a lack of correlation between body size and the segmentation period in various species of mammals. The work supports their findings, and it would be good to see this discussed in the text.

We are not certain how best to compare our *in-vivo* results in externally developing fish embryos to *in-vitro* mammalian 2-D cell cultures. In our view, the correlation of embryo size, larval and adult size that we find in *Oryzias* might not necessarily hold in mammalian species, which would make a comparison more difficult. We do cite the work mentioned so the reader is pointed towards this interesting, complementary literature.

Significance

The work is quite remarkable in terms of the multigenerational genetic analysis performed. The authors have analysed >600 embryos from three separate generations to obtain quantitative data to answer their question

Full Revision

(herculean task!). Moreover, they have associated this characterization to specific SNPs. Then, to go beyond the association, they have generated mutant lines and identified specific genes associated to the traits they set out to decipher.

To my knowledge, this is the first project that aims to identify the genetic determinants for developmental timing. Recent work on developmental timing in mammals has focused on interspecies comparisons and does not provide genetic evidence or insight into how tempo is regulated in the genome. As for vertebrates, recent work from zebrafish has profiled temperature effects on cell proportions and developmental timing. However, the genetic approach of this work is quite elegant and neat.

Conceptually, it is quite important and unexpected that overall size and tempo are not related. Body size, lifespan, basal metabolic rates and gestational period correlate positively and we tend to think that mechanistically they would all be connected to one another. This paper and Lazaro et al. 2023 (ref 27) are one of the first in which this preconception is challenged in a very methodical and conclusive manner. I believe the work is a breakthrough for the field and this work would be interesting for the field of biological timing, for the segmentation clock community and more broadly for all developmental biologists.

My field is quantitative stem cell biology and I work on developmental timing myself, so I acknowledge that I am biased in the enthusiasm for the work. It should be noted that as an expert on the field, I have identified instances where other work hasn't been as insightful or well developed in comparison to this piece. It is also worth noting that I am not an expert in fish development, phylogenetic studies or GWAS analyses, so I am not capable to assess any pitfalls in that respect.

Reviewer #3 (Evidence, reproducibility and clarity (Required)):

Summary:

This manuscript explores the temporal and spatial regulation of vertebrate body axis development and patterning. In the early stages of vertebrate embryo development, the axial mesoderm (presomitic mesoderm - PSM) undergoes segmentation, forming structures known as somites. The exact genetic regulation governing somite and PSM size, and their relationship to the periodicity of somite formation remains unclear.

To address this, the authors used two evolutionarily closely related Medaka species, *Oryzias sakaizumii* and *Oryzias latipes*, which, although having distinct characteristics, can produce viable offspring. Through analysis spanning parental (generation F0) and offspring (generations F1 and F2) generations, the authors observed a correlation between PSM and somite size. However, they found that size scaling does not correlate with the timing of somitogenesis.

Furthermore, employing developmental quantitative trait loci (devQTL) mapping, the authors identified several new candidate loci that may play a role during somitogenesis, influencing timing of segment formation or segment size. The significance of these loci was confirmed through an innovative CRISPR-Cas9 gene editing approach.

This study highlights that the spatial and temporal aspects of vertebrate segmentation are independently controlled by distinct genetic modular mechanisms.

Major comments:

Full Revision

1) In the main text page 3, lines 11 and 12, the authors state that the periodicity of the embryo clock of the F1 generation is the intermediate between the parental F0 lineages. However, the authors look only at the periodicity of the Cab strain (*Oryzias latipes*) segmentation clock. The authors should have a reporter fish line for the Kaga strain (*Oryzias sakaizumii*) to compare the segmentation clock of both parental strains and their offspring. Since it could be time consuming and laborious, I advise to alternatively rephrase the text of the manuscript.

We agree a careful distinction between segment forming rate (measured based on morphology) and clock period (measured using the novel reporter we generated) is essential. We show that both measures correlate very well in Cab, in both F0 and F1 and F2 carrying the Cab allele. For Kaga F0, we indeed can only provide the rate of somite formation, which nevertheless allows comparison due to the strong correlation to the clock period we have found. We have rephrased the text accordingly.

2) It is evident that only a few F0 and F1 animals were analyzed in comparison with the F2 generation. Could the authors kindly explain whether and how this could bias or skew the observed results?

We provide statistical evidence through the F-test of equality that the variances between the F0, F1 and F2 samples are equal. Additionally if we sub-sample and separate the F2 data into groups of 100 embryos (instead of all 638) we get the same distribution of the F2s. We therefore believe that this is sufficient evidence against a bias or skew in the results.

3) It would be interesting to create fish lines with the validated CRISPR-Cas9 gene manipulations in different genetic contexts (Cab or Kaga) to analyze the true impact on the segmentation clock and/or PSM & somite sizes.

We agree with the reviewer this would in principle be of interest indeed, please see our response to reviewer 1 earlier.

4) Please add the results of the Go Analysis as supplementary material.
We have added the GO analysis in new Supplementary Figure 7E.

Minor comments:

1) In the main text, page 2, line 29, Supplementary Figure 1D should be referenced.

We have added a clearer phylogeny and geographical location of the different species in new Figure 1 A-B. And reference it at the requested location.

2) In the main text, page 2, line 32, the authors refer to Figure 1B, but it should be 1C.

We have corrected the information.

3) Regarding the topic "Correlation of segmentation timing and size in the *Oryzias* genus" the authors should also give information on the total time of development of the different *Oryzias* species, as well as the total number of formed somites.

Full Revision

We follow this recommendation and have added this information in new Supplementary Figure 5. We also now include segment number measured in F2 embryos. We indeed view segmentation rate as a proxy for developmental rate, which however needs to be distinguished from total developmental time. The latter can be measured for instance by quantifying hatching time, which we did. These measurements show that Kaga, Cab and *O.hubbsi* embryos kept at constant 28 degrees started hatching on the same day while *O.minutillus* and *O.mekongensis* embryos started hatching one day earlier. We have not included this data in the manuscript because we think a distinction should be made between rate of development and total development time.

4) In Figures 3A and B, please add info on the F1 lines for comparison.

The information on F1 lines is provided in Supplementary Figure 3

5) Supplementary Figures 2F shows that the generation F1 PSM is similar to Cab F0, and not an intermediate between Kaga F0 and Cab F0. This is interesting and should be discussed.

We show that the F1 PSM is indeed closer to the PSM of Cab than it is to the Kaga PSM. This is indeed intriguing and we have now commented on this point directly in the text.

6) Supplementary Figures 6C to H are not mentioned either in the main text or in the extended information. Please add/mention accordingly.

We have added references to both in the text

7) The order of Supplementary Figure 8 E to H and A to D appears to be not correct and not following the flow of the text. Please update/correct accordingly.

We have updated the text accordingly.

8) The authors should choose between "Fig.", "Fig", "fig.", "fig" or "Figure". All 'variants' can be found in the text. Noted, and updated. Fig. is used for main figures and fig. is used for supplementary figures.

9) The color scheme of several figures (graphs with colored dots) should be revised. Several appear to be difficult to discern and analyze.

We have enhanced the colours and increased the font on the figure panels. The colour panel was chosen to be colour-blind friendly.

10) Please address/discuss following questions: What are the known somitogenesis regulating genes in Medaka? How do they correlate with the new candidates?

The candidates we found and tested had not been implicated in regulating the tempo of segmentation or PSM size, while for some a role in somite formation had been previously established, hence the enrichment in GO analysis Somitogenesis.

Full Revision

Reviewer #3 (Significance (Required)):

General assessment:

This interesting manuscript describes a novel approach to study and find new players relevant to the regulation of vertebrate segmentation. By employing this innovative methodology, the authors could elegantly demonstrate that the segmentation clock periodicity is independent from the sizes of the PSM and forming somites. The authors were further able to find new genes that may be involved in the regulation of the segmentation clock periodicity and/or the size of the PSM & somites. A limitation of this study is the fact that the results mainly rely on differences between the two species. The integration of additional Medaka species would be beneficial and may help uncover relevant genes and genetic contexts.

Advance:

To my best knowledge this is the first time that such a methodology was employed to study the segmentation clock and axial development. Although the topic has been extensively studied in several model organisms, such as mice, chicken, and zebrafish, none of them correlated the size of the embryonic tissues and the periodicity of the embryo clock. This study brings novel technological and functional advances to the study of vertebrate axial development.

Audience:

This work is particularly interesting to basic researchers, especially in the field of developmental biology and represents a fresh new approach to study a core developmental process. This study further opens the exciting possibility of using a similar methodology to investigate other aspects of vertebrate development. It is a timely and important manuscript which could be of interest to a wider scientific audience and readership.

Dear Alexander,

Thank you for submitting your revised Review Commons manuscript to The EMBO Journal. I sincerely apologise for the protracted assessment process due to delays in referee comment submission. Your study has now been seen by two of the original referees, who now find that most of their previous concerns have been addressed and recommend acceptance of the manuscript.

In addition to the final minor points raised by the reviewer #1, there are a few editorial points that need addressing before I can extend acceptance of the manuscript:

1. Please submit up to five keywords.
2. Please make sure that the order of the sections in the manuscript is as follows: abstract, introduction, results, discussion, materials & methods, data availability section, acknowledgments, disclosure statement and competing interests, references, main figure legends, tables, expanded figure legends.
3. Please add full funding information in the "Acknowledgements" section.
4. Please submit a complete author checklist, which you can download from our author guidelines (<https://www.embopress.org/pb-assets/embo-site/EMBO%20Press%20Author%20Checklist-1642513524327.xlsx>). Please insert information in the checklist that is also reflected in the manuscript. The completed author checklist will also be part of the Review Process File.
5. We are missing the ORCID iD for the co-corresponding author Ewan Birney. In order to link the ORCID iD to the account in our manuscript tracking system, the author in question has to do the following:
 - Click the 'Modify Profile' link at the bottom of your homepage in our system.
 - On the next page you will see a box halfway down the page titled ORCID*. Below this box is red text reading 'To Register/Link to ORCID, click here'. Please follow that link: you will be taken to ORCID where you can log in to your account (or create an account if you don't have one)
 - You will then be asked to authorise Wiley to access your ORCID information. Once you have approved the linking, you will be brought back to our manuscript system.Unfortunately, we cannot do this linking on the author's behalf for security reasons.
6. There are 10 supplementary figures. Up to 5 can be made EV (Expanded View) figures and should be uploaded as individual high resolution figure files. Their legends should be in the manuscript, after the main figure legends, under the heading "Expanded View Figure Legends". Their nomenclature should be updated to "Figure EV1" - "Figure EV5". The remaining supplementary figures should be compiled in a PDF, with their legends added underneath each figure, and named "Appendix Figure S1" etc. The PDF should be labelled "Appendix" and needs a table of contents with page numbers. Further information on the format is available here: <https://www.embopress.org/page/journal/14602075/authorguide#expandedview>.
7. Please upload the main and EV figures as individual production quality figure files in the .eps, .tif, or .jpg format (one file per figure).
8. There are references to Data S1-S3, but such files are not provided. Please correct.
9. CRediT has replaced the traditional author contributions section because it offers a systematic, machine-readable author contributions format that allows for more effective research assessment. Please remove the Authors Contributions from the manuscript and use the free text boxes beneath each contributing author's name in our online submission system to add specific details on the author's contribution. More information is available in our guide to authors.
10. Please rename "Conflict of interest" section into "Disclosure and competing interests statement" (further info: <https://www.embopress.org/page/journal/14602075/authorguide#conflictsofinterest>).
11. Please update references according to The EMBO Journal style - where there are more than 10 authors on a paper, the first 10 should be listed, followed by 'et al.' Please see further information here: <https://www.embopress.org/page/journal/14602075/authorguide#referencesformat>
12. In our standard image integrity check, we noticed reuse of figure panels between supplementary figure 8. F' and H', and supplementary figure 10 C, G and H (control). Please indicate in the figure legends that these images were derived from the same sample.
13. Our data editors have flagged the following issues in figure legends that need correcting:
 - Please indicate the statistical test used for data analysis in the legend of figure 4a.
 - Please add information on the number and nature of replicates in the legends of figures 1c-d.
 - Please define the error bars in the legend of figure 4b.
 - Please define the measure of center for the error bars in the legends of figures 2d-e; 4d.
14. At EMBO Press we ask authors to provide source data for the main manuscript figures. Our source data coordinator will contact you to discuss which figure panels we would need source data for and will also provide you with helpful tips on how to upload and organize the files.
15. Papers published in The EMBO Journal are accompanied online by a 'Synopsis' to enhance discoverability of the manuscript. It consists of A) a short (1-2 sentences) summary of the findings and their significance, B) 3-4 bullet points highlighting key results and C) a synopsis image that is 550x300-600 pixels large (width x height, jpeg or png format). You can either show a model or key data in the synopsis image. Please note that the image size is rather small and that text needs to be readable at the final size. Please send us this information together with the revised manuscript.

With best wishes,

Ieva

Revision to The EMBO Journal should be submitted online within 90 days, unless an extension has been requested and approved by the editor; please click on the link below to submit the revision online before 29th Aug 2024:

Link Not Available

Referee #1:

The authors have now addressed all points raised by the referees, and the study should be published.

I would recommend the authors to include as as main the new data shown in new Supplementary figure 7 and new Supplementary figure 9. Fig S7 justifies the rationale for candidate prioritization and go analysis of the devQTL peaks. Fig S9 describes the characterization that has been done for mespb and pcdh10b mutants.

Reference 27 is cited alongside two other references (25 and 26) in the introduction, but it does not indicate that the work provides findings on the relationship between tempo and tissue size. While reference 27 presents in vitro measures for the period, there are also comparisons with whole organismal measures (Fig 2A), and the results very much resemble the findings in this manuscript. This publication is the closest one to the work presented here, so it would be relevant for the authors to put their work and ref 27 in perspective.

Referee #2:

The authors have satisfactorily addressed all my concerns. I have no further comments and strongly recommend publication of this manuscript.

Rev_Com_number: RC-2023-02279
New_manu_number: EMBOJ-2024-117664-T
Corr_author: Aulehla
Title: Modular control of time and space during vertebrate axis segmentation

Referee #1:

The authors have now addressed all points raised by the referees, and the study should be published.

I would recommend the authors to include as main the new data shown in new Supplementary figure 7 and new Supplementary figure 9. Fig S7 justifies the rationale for candidate prioritization and go analysis of the devQTL peaks. Fig S9 describes the characterization that has been done for *mespb* and *pcdh10b* mutants.

We have taken this into account and have now add both figures as Expanded View as EV4 and EV5.

Reference 27 is cited alongside two other references (25 and 26) in the introduction, but it does not indicate that the work provides findings on the relationship between tempo and tissue size. While reference 27 presents *in vitro* measures for the period, there are also comparisons with whole organismal measures (Fig 2A), and the results very much resemble the findings in this manuscript. This publication is the closest one to the work presented here, so it would be relevant for the authors to put their work and ref 27 in perspective.

Reference 27 is indeed a very interesting and relevant manuscript. It clearly shows that the segmentation clock does not correlate with metabolic activity in *in-vitro* settings across different mammalian cell lines but instead correlates quite well with length of embryogenesis. While the authors do not show any tissue or embryonic size measurements in their samples, the lack of correlation to adult size is indeed intriguing. It is important however to distinguish between embryonic and adult sizes. We have added a sentence to reflect that in our discussion.

Referee #2:

The authors have satisfactorily addressed all my concerns. I have no further comments and strongly recommend publication of this manuscript.

We thank the reviewer for his/her comments.

Dear Alexander and Ali,

Thank you for addressing the final formatting issues. I sincerely apologise for the slow process from our side due to the high number of submissions that we experience at the moment, combined with conference travel last week. I am now pleased to inform you that your manuscript has been accepted for publication - congratulations on a nice study!

Before we forward your manuscript to our publishers, I would like to propose some minor edits in the manuscript title, abstract and synopsis (please see below and the attached manuscript text file). These are mainly aimed at increasing the accessibility of the study to our more general readership. I have also written a short blurb that will accompany the title of your manuscript in our online system. Please let me know if any corrections or adjustments are needed:

Title:

Modular control of timing and segment size scaling during vertebrate axis segmentation

Blurb:

Interspecies crosses in rice fish (*Orzias* spp.) show that both the rate of development and the final size of segments increase in proportion to fish size but are regulated by independent mechanisms.

Synopsis:

The molecular relationships that link developmental timing and organismal size currently remain underexplored. Here, a cross-species analysis in *Oryzias* fish measures the timing of embryonic body axis segmentation and overall organismal size, providing evidence that timing and size control are dissociable modules.

- Real-time imaging of segmentation clock in F2 embryos from interspecies crosses of *Oryzias* fish reveals a broad phenotypic spectrum.
- F2 embryos exhibit a linear relationship between PSM and somite size, that a developmental constraint mechanism underlies spatial scaling.
- Expression quantitative trait loci (eQTL) analysis of F2 embryos identifies distinct genomic loci correlated with the segmentation timing and tissue size, respectively.

If you have any questions, please do not hesitate to contact the Editorial Office. Thank you for this contribution to The EMBO Journal and congratulations on a great paper!

With best wishes,

leva

leva Gailite, PhD
Senior Scientific Editor
The EMBO Journal
Meyerohofstrasse 1
D-69117 Heidelberg
Tel: +4962218891309
i.gailite@embojournal.org

>>> Please note that it is The EMBO Journal policy for the transcript of the editorial process (containing referee reports and your response letter) to be published as an online supplement to each paper. If you do NOT want this, you will need to inform the

Editorial Office via email immediately. More information is available here: https://www.embopress.org/transparent-process#Review_Process

Rev_Com_number: RC-2023-02279

New_manu_number: EMBOJ-2024-117664R

Corr_author: Aulehla

Title: Modular control of time and space during vertebrate axis segmentation